# Atmospheric triggering conditions and climatic disposition of landslides in Kyrgyzstan and Tajikistan at the beginning of the 21st century

Xun Wang[1], Marco Otto[1], and Dieter Scherer[1]

[1]Chair of Climatology, Technische Universität Berlin, Berlin, 12165, Germany

**Correspondence:** Xun Wang (xun.wang@tu-berlin.de)

**Abstract.** Landslide is a major natural hazard in Kyrgyzstan and Tajikistan. Knowledge about atmospheric triggering conditions and climatic disposition of landslides in Kyrgyzstan and Tajikistan is limited, even though this topic has already been investigated thoroughly in other parts of the world. In this study, the newly developed, high-resolution High Asia Refined Analysis version 2 (HAR v2) data set generated by dynamical downscaling was combined with historical landslide inventories to analyze the atmospheric conditions that initialized landslides in Kyrgyzstan and Tajikistan. The results indicate the crucial role of snowmelt in landslide triggering processes since it contributes to the initialization of 40% of landslide events. Objective thresholds for rainfall, snowmelt, as well as the sum of rainfall and snowmelt (rainfall+snowmelt) were defined. Thresholds defined by rainfall+snowmelt have the best predictive performance. Mean intensity, peak intensity, and the accumulated amount of rainfall+snowmelt events show similar predictive performance. Using the entire period of rainfall+snowmelt events results in better predictive performance than just considering the period up-to landslide occurrence. Mean annual exceedance maps were derived from defined regional thresholds for rainfall+snowmelt. Mean annual exceedance maps depict climatic disposition and have added value in landslide susceptibility mapping. The results reported in this study highlight the potential of dynamical downscaling products generated by regional climate models in landslide prediction.

## 1 Introduction

Landslide is one of the most severe natural hazards in Kyrgyzstan and Tajikistan. More than 300 big landslides occurred in Kyrgyzstan from 1993 to 2010, causing 256 fatalities and direct economic losses of 2.5 million USD per year (Torgoev et al., 2012). Under global warming, wildfires, glacial retreat, and permafrost degradation are much likely to enhance slope instabilities in mountainous areas (Froude and Petley, 2018; Palmer, 2020), making these regions, including Kyrgyzstan and Tajikistan, more vulnerable to climate change. The occurrence of landslides depends on disposition and triggering events. Disposition refers to the general settings that make slopes prone to failure without actually initiating it, such as slope gradient and aspect, geology, vegetation cover, climate, etc. (Dai et al., 2002). Common triggers for landslides are extreme and prolonged rainfall, rapid snowmelt, as well as earthquakes (Wieczorek, 1996).

The majority of landslide research in Kyrgyzstan and Tajikistan focused on characterizing landslide susceptibility, i.e., "where" landslides are prone to occur (e.g., Braun et al., 2015; Saponaro et al., 2015; Havenith et al., 2015b), and how

to improve the landslide susceptibility models (Ozturk et al., 2020; Barbosa et al., 2021). But little attention is paid to the atmospheric triggering conditions, and our knowledge of "when" landslides are likely to occur is limited in this region. In addition, most landslide susceptibility studies only took non-climatic factors into account or simply applied annual precipitation as a climatic factor. According to Segoni et al. (2018), no rainfall threshold for landslide triggering has been defined for Kyrgyzstan and Tajikistan yet, even though this topic has already been thoroughly investigated in other parts of the world with high landslide susceptibility (e.g., Berti et al., 2012; Gariano et al., 2015; Giannecchini et al., 2016; Leonarduzzi et al., 2017). The reasons are twofold. Firstly, although landslide inventories have been developed in this region, e.g., the Tien Shan Geohazards Database (Havenith et al., 2015a, b) and the multi-temporal landslide inventory from Behling and Roessner (2020), there is a lack of landslide inventories with the exact date of landslide occurrence. Given the highly dynamic nature of weather phenomena, at least a daily timestamp of landslide records is required to investigate weather conditions that trigger landslides. Secondly, there is a lack of atmospheric data. The number of in-situ observation stations in Kyrgyzstan and Tajikistan decreased sharply in the 1990s due to reduced funding. There are currently eight stations in Kyrgyzstan and 26 stations in Tajikistan available from Global Surface Summary of the Day (GSOD), which is a publicly available data set. These numbers are already significantly below the recommendation of the World Meteorological Organization, even for flat areas (Ilyasov et al., 2013). Despite the sparse distribution, most GSOD stations are located in low-lying valleys and are not fully representative of the area.

Rainfall is the most common trigger of landslide all over the world (Wieczorek, 1996). Over snow-covered regions, snowmelt is recognized as another common trigger of shallow landslides and debris flows (Wieczorek, 1996; Mostbauer et al., 2018). In Kyrgyzstan and Tajikistan, more than half of the annual precipitation falls in the form of snow. Snow cover duration over high mountain ranges in the Tien Shan and the Pamir is more than 200 days per year (Dietz et al., 2014). A large amount of water stored in snowpacks is released during the melting season. Snowmelt is another important source of water infiltrating into the soil that increases slope instability. Thus, in Kyrgyzstan and Tajikistan, snowmelt might also play a role in landslide triggering besides rainfall. But snowmelt is not as easy to be observed as rainfall and might often be neglected as a landslide trigger, especially when co-occurring with rainfall.

There are two main approaches to assess rainfall thresholds for landslide triggering. The first approach is physically based and requires detailed lithological, morphological, and geotechnical information of each landslide event (Guzzetti et al., 2007). Unfortunately, this level of detail is usually restricted to small areas and is not available for the whole of Kyrgyzstan and Tajikistan. The second one is the empirical approach based on historical landslide and rainfall data. The majority of studies applying this approach relied on rain gauge data to analyze rainfall thresholds (e.g., Berti et al., 2012; Khan et al., 2012; Bui et al., 2013). However, rain gauge data are point measurements that cannot capture the large spatial heterogeneity of rainfall, especially over complex terrains. Gridded products can provide continuous data in both space and time and can be used in detecting atmospheric triggering conditions of landslides.

We aim to analyze the atmospheric triggering conditions of landslides and generate climatic disposition maps that contain information on these triggering conditions in Kyrgyzstan and Tajikistan. For this purpose, we combined freely available gridded atmospheric data with historical landslide events. Atmospheric triggers for each landslide event were determined by the co-occurrence of landslide and weather events. Properties (mean intensity, peak intensity, accumulated amount) of landslide

triggering events and non-landslide triggering events were compared. Objective thresholds of these properties for different atmospheric triggers (rainfall, snowmelt, and the sum of rainfall and snowmelt) were defined so that they can best separate the atmospheric conditions that resulted and did not result in landslides. Finally, we applied the thresholds with the best predictive performance to generate maps of mean annual exceedance. In this way, we can transform the weather-scale triggering conditions into climate-scale dispositions (hereafter referred to as "climatic disposition").

The objective of this study is threefold: (1) investigate the role of snowmelt in landslide triggering processes; (2) find appropriate quantities of atmospheric triggers for assessing landslide hazards; (3) characterize climatic disposition in terms of rainfall and snowmelt over Kyrgyzstan and Tajikistan.

The paper is organized as follows: we describe the data and methods used in this study in the following section. Results are presented in section 3 and discussed in 4. Conclusions are drawn in section 5.

## 2 Data and method

### 2.1 Data

#### 2.1.1 Landslide catalog

Landslide events used in this study come from two sources: the Global Landslide Catalog (GLC) (Kirschbaum et al., 2010, 2015) and the Global Fatal Landslide Database (GFLD) (Froude and Petley, 2018). GLC has been compiled by NASA since 2007 and contains all types of mass movements triggered mostly by rainfall. The sources of the GLC are mainly media reports, disaster databases, and scientific reports. The GFLD only includes landslide events that caused fatality obtained from media reports. It currently covers the period from 2004 to 2017. These two landslide inventories were chosen because, to the best of our knowledge, they are the only ones with the exact landslide dates available for the study region.

We selected landslide events triggered by atmospheric factors in Kyrgyzstan and Tajikistan from 2007-2018 from the GLC and 2004-2017 from the GFLD. Then we merged these two data sets and deleted duplicated events that occurred on the same day and came from the same source link, resulting in 96 landslide events for Kyrgyzstan and Tajikistan from 2004 to 2018 (Fig. 1).

#### 2.1.2 Atmospheric data

Rainfall and snowmelt data are extracted from the HAR v2. The HAR v2 is a newly developed regional atmospheric data set. It was generated by dynamical downscaling of the ERA5 reanalysis data using the Weather Research and Forecasting model (WRF). It provides atmospheric data with high resolution and accuracy over High Mountain Asia (Hamm et al., 2020; Wang et al., 2021). Detailed modeling strategies of the HAR v2 are described in Wang et al. (2021). The HAR v2 has a grid spacing of 10 km and is available in hourly, daily, monthly and yearly aggregations. Daily products were used in this study to determine the climatic trigger of each landslide event (section 2.2.1) and to define thresholds for landslide triggering (section 2.2.2). Rainfall was calculated as the difference between total precipitation and snowfall. Snowmelt is not a standard output of

the WRF and was calculated using the Surface Energy Balance (SEB). The SEB in the HAR v2 is resolved by the Noah Land Surface Model (LSM) (Tewari et al., 2004):

$$H_m = R_n - H_s - H_l - H_g \tag{1}$$

where $R_n$, $H_s$, $H_l$ and $H_g$ are net radiation, sensible heat flux, latent heat flux, and ground heat flux in $\mathrm{W\,m^{-2}}$, respectively. These four variables are directly available in the HAR v2. $H_m$ is the heat flux for melting and refreezing in $\mathrm{W\,m^{-2}}$. $H_m > 0$ indicates melting process, while $H_m < 0$ refers to refreezing process. When $H_m > 0$, snowmelt $h_m$ $(\mathrm{kg\,m^{-2}\,s^{-1}})$ is calculated as:

$$h_m = H_m/\lambda_m \tag{2}$$

where $\lambda_m$ is the latent heat of fusion. When the calculated $h_m$ is greater than snow water equivalent, then $h_m$ is set to be equal to snow water equivalent.

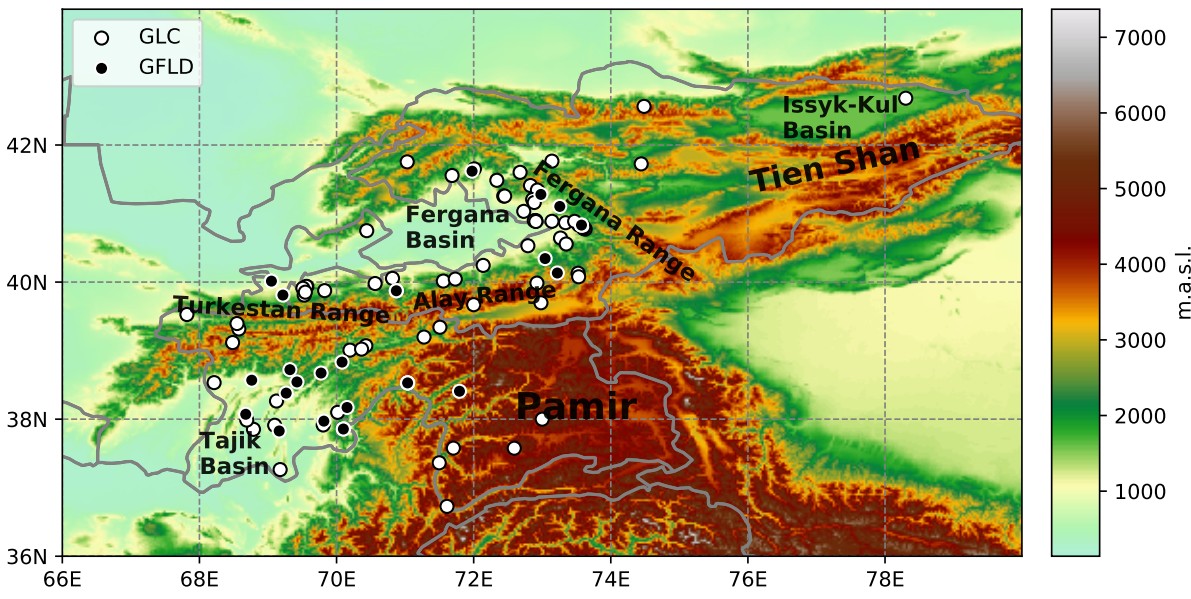

**Figure 1.** Landslide events from 2004-2018 extracted from the GLC (white points) and the GFLD (black points). Background contour is topography from Digital Elevation Model (DEM) data from Shuttle Radar Topographic Mission (SRTM).

## 2.2 Methods

### 2.2.1 Determine the atmospheric trigger of landslide events

The atmospheric trigger of a landslide event is determined by the co-occurrence of the landslide event with rainfall and snowmelt event. If a landslide event only occurred within or one day after a rainfall (snowmelt) event, then this landslide event is defined as rainfall (snowmelt) triggered. If there are both a rainfall event and a snowmelt event on the day or one day before the landslide occurrence day, then the atmospheric trigger of this landslide event is mixed.

To define a rainfall (snowmelt) event, the daily time series of rainfall(snowmelt) were extracted from the grid cells where landslides occurred. For each time series, an independent rainfall (snowmelt) event is defined as a series of consecutive days in which more than $0.2\,\mathrm{mm\,d^{-1}}$ of rainfall (snowmelt) is simulated. The value of $0.2\,\mathrm{mm\,d^{-1}}$ is chosen because it is the traditional precision of daily precipitation measurement (Jarraud, 2008) and can be applied to separate dry and wet conditions (Rodwell et al., 2010).

### 2.2.2 Threshold model for atmospheric triggers

The threshold model developed in this study contains three steps: (1) define landslide triggering events and non-triggering events; (2) define the thresholds for rainfall, snowmelt, and the sum of rainfall and snowmelt (hereafter referred to as rainfall+snowmelt) based on maximizing the predictive performance using $2 \times 2$ contingency tables; (3) validate and assess the uncertainties of the defined thresholds. The methods for the first two steps were adopted from Leonarduzzi et al. (2017). Only the landslide events, the climatic triggers of which could be determined, were used for threshold modeling.

The first step is to define landslide triggering events and non-triggering events for rainfall, snowmelt, and rainfall+snowmelt. Here, we take rainfall as an example to describe the procedure. First, the method used in section 2.2.1 is applied to define rainfall events for each time series extracted from grid cells where landslides occurred. Next, if a landslide event occurred during or one day after a rainfall event, then this rainfall event is classified as a landslide triggering event (LTE). Given the uncertainty in timestamps of landslide events, the day after is also considered as a temporal relaxation. Otherwise, if a rainfall event is not associated with any landslide events, it is classified as a non-landslide triggering event (NLTE). For each rainfall event, we calculated three event properties: mean intensity $I_{mean}$, maximum intensity $I_{max}$, and the accumulated amount of rainfall for the entire event $Q$. For triggering events, we also calculated these three properties by only considering the period up to the day of the landslide occurrence (hereafter referred to as UTL, meaning Up-To-Landslide). Note that, not all the landslide events co-occurred with a rainfall event. For these events, we set $I_{mean}$, $I_{max}$, and $Q$ to zero. The same procedure for defining LTEs and NLTEs was conducted for snowmelt and rainfall+snowmelt as well.

The second step is to define thresholds of rainfall, snowmelt, and rainfall+snowmelt for entire events and UTL events, using $I_{mean}$, $I_{max}$, and $Q$. No single threshold can perfectly separate LTEs from NLTEs since their distributions overlap. We applied $2 \times 2$ contingency tables to select the threshold that yields the best predictive performance. Using a certain threshold as a binary classifier, LTEs and NLTEs were categorized into true positive (TP), true negative (TN), false positive (FP), and false negative (FN). The Peirce Skill Score (PSS) (Hanssen and Kuipers, 1965) was applied as the measure of the predictive

performance because it is trail-independent, which means it is unbiased even when the numbers of LTEs and NLTEs are not equally presented (Woodcock, 1976). The PSS is also known as the Hanssen-Kuiper skill score and the true skill statistic. It is calculated as the difference between Hit Rate (HR) and False Alarm Rate (FAR):

$$PSS = HR - FAR \tag{3}$$

$$HR = \frac{TP}{TP + FN} \tag{4}$$

$$FAR = \frac{FP}{FP + TN} \tag{5}$$

We chose the threshold that maximizes the PSS. We also computed the Euclidean distance ($d$) to the optimal point (HR=1, FAR=0), which is another commonly used skill score in this application (e.g., Gariano et al., 2015; Piciullo et al., 2017; Postance et al., 2018; Zhuo et al., 2019). Additionally, the receiver operating characteristic (ROC) curve was used to determine the general predictive power of a certain predictor by calculating the area under the ROC curve (AUC) (Fawcett, 2006).

The last step is to validate the threshold model and assess uncertainty. For the calibration of thresholds, all landslide event samples were utilized, and corresponding statistic measures were calculated, i.e., the threshold model was trained and tested on the same data set. To test the model's predictive ability on an unseen data set, we performed k-fold cross-validation. Landslide events were randomly split into k folds with k=8. Then for each unique fold, the fold was taken as the testing set, and the remaining k-1 folds were taken as the training set. Mean values of thresholds, the corresponding statistic measures, as well as their uncertainties represented by standard deviations were reported.

### 2.2.3 Mean annual exceedance

Mean annual exceedance ($\overline{N}_{th}$) is calculated for each HAR v2 grid cell. It is defined as the number of events that exceed a certain threshold over a certain period ($N_{th}$) divided by the total number of years ($N_a$):

$$\overline{N}_{th} = \frac{N_{th}}{N_a} \tag{6}$$

The unit of $\overline{N}_{th}$ is the number of events per year. Mean annual exceedance transforms weather-scale triggering conditions to climate-scale disposition. It depicts where landslides are likely to occur from the climatic aspect.

## 3 Results

### 3.1 The role of snowmelt in landslide triggering

Figure 2 shows the climatology of seasonal rainfall, snowmelt, and rainfall+snowmelt resolved by the HAR v2. We define seasons as commonly done in meteorology, spanning three months each: winter (December-February, DJF), spring (March-May, MAM), summer (June-August, JJA), and autumn (September-November, SON). A high amount of rainfall concentrates in the western foothill of the Fergana Range, the northern foothill of the Turkestan Range, and the Tajik Basin in spring and shifts northeastwards into the Tien Shan in summer. Snowmelt occurs in spring over most high elevated areas. In summer,

while most regions are snowmelt-free, the Pamir plateau still experiences a high amount of continuous snowmelt, which is in line with the results by Dietz et al. (2014) using remote sensing data.

Atmospheric triggers for each landslide event are determined using the method described in section 2.2.1, and the results are shown in Fig. 3. Table A1 lists all 96 events and the climatic triggers detected by the HAR v2. Figure A1 shows the temporal process of rainfall and snowmelt for selected landslide cases. Nine landslide events did not occur within any rainfall event, snowmelt event, or rainfall+snowmelt event. This mismatch between landslide information and weather information stems from the uncertainties in landslide locations and timing, as well as the uncertainties from rainfall and snowmelt simulated in

the HAR v2 (detailed discussion in section 4.1). These nine events are referred to as "not detected" (white points in Fig. 3) and are excluded. The remaining 87 landslide events were used for further analysis. Landslide events that were only triggered by rainfall mainly cluster in Tajik Basin and the northeastern rim of the Fergana Basin, where the contribution of rainfall to the annual sum of rainfall and snowmelt is high (Fig. 3).

The annual cycles of rainfall, snowmelt, and rainfall+snowmelt are compared with monthly landslide occurrences in Fig.

4. The study region experiences a peak of landslide activity in April and May, which corresponds with the peak of rainfall+snowmelt. While rainfall is the dominant trigger of landslides, snowmelt contributes to triggering 40% of landslide events (35 out of 87). There are 29% of landslide events (25 out of 87) that are attributed to the combined effect of rainfall and snowmelt. Most snowmelt-contributing events occurred in April when snowmelt amount is the highest. March and June have almost the same amount of rainfall+snowmelt. However, there are more landslide occurrences in June. This could be resulted

from still frozen soil in March, which stabilizes the slope. As shown in Fig. 4a, both soil temperature at the top soil layer (0-0.1m) and air temperature at $2\,\mathrm{m}$ are still below zero in March.

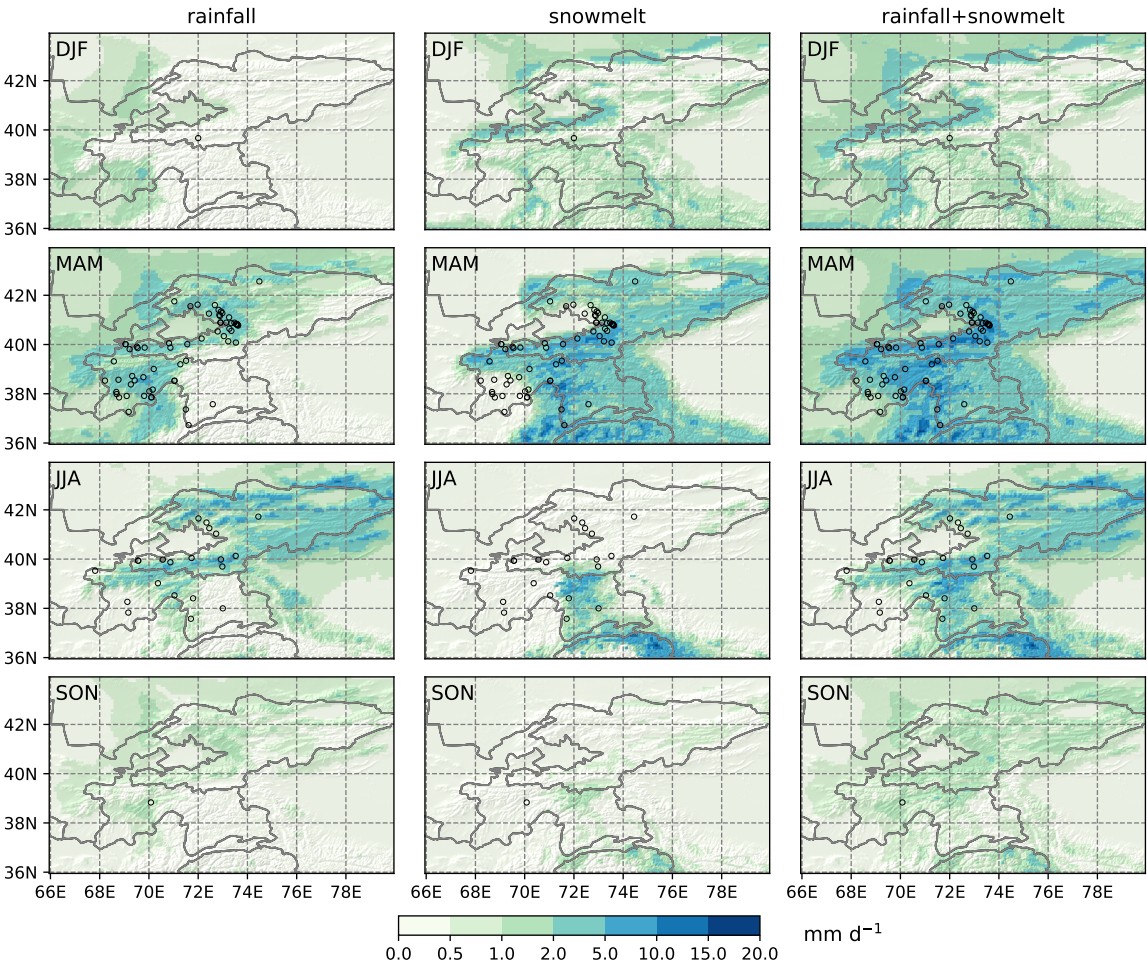

**Figure 2.** Seasonal rainfall, snowmelt, and rainfall+snowmelt from the HAR v2 from 2004-2018. Black circles: seasonal landslide events from GLC and GFLD. Topographic shading is based on DEM data from SRTM. DJF: December-February; MAM: March-May; JJA: June-August; SON: September-November.

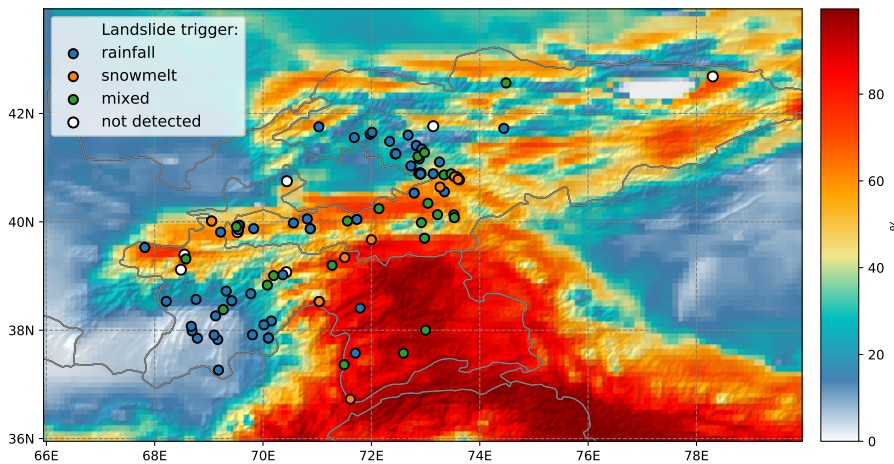

**Figure 3.** Contribution (%) of snowmelt to annual sum of rainfall and snowmelt (background contour) and climatic triggers of 96 landslide events extracted from GLC and GFLD (points). Topographic shading is based on DEM data from SRTM.

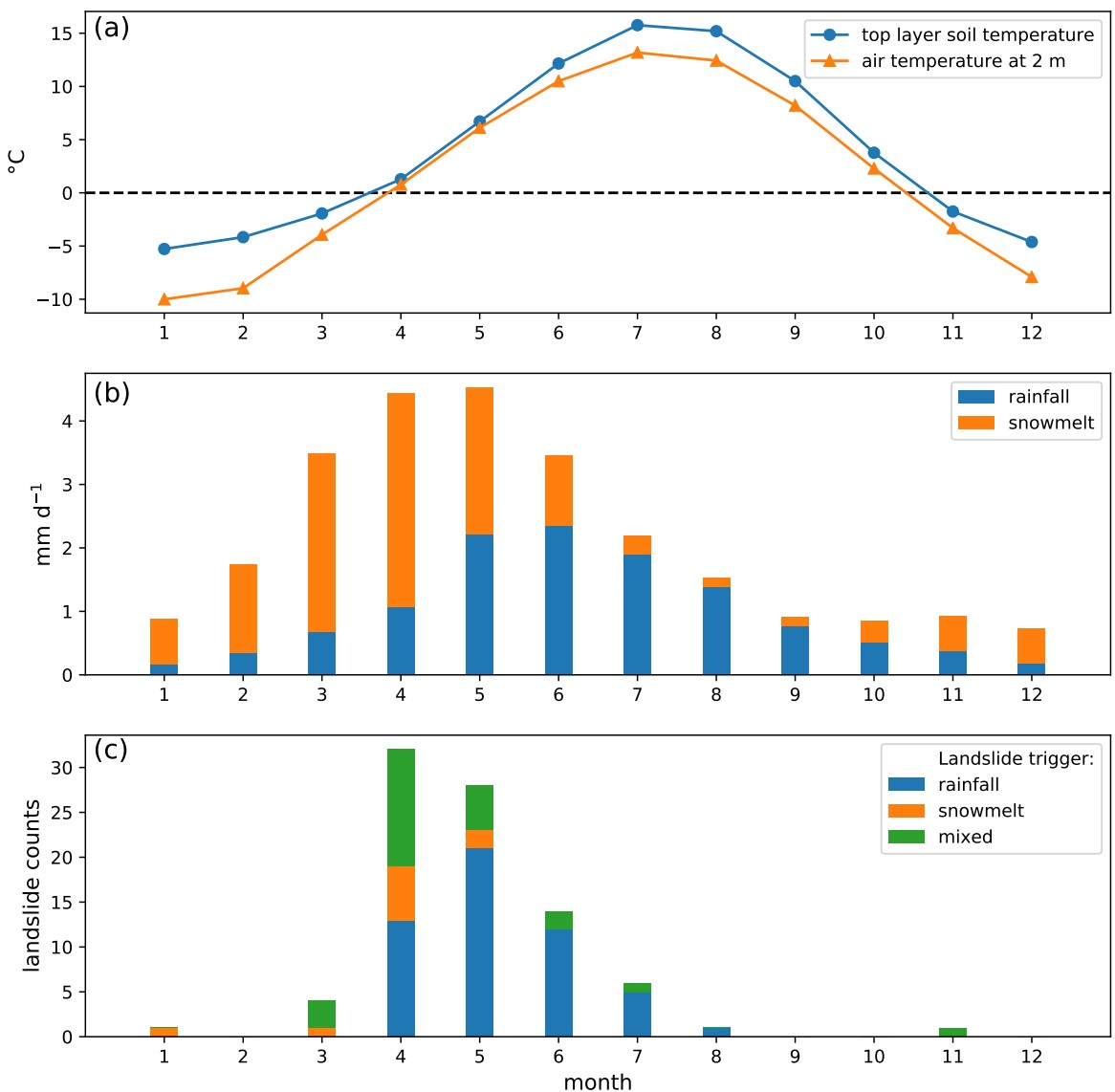

**Figure 4.** (a) Mean monthly soil temperature at the top soil layer (0-0.1m) and air temperature at 2 m averaged over Kyrgyzstan and Tajikistan extracted from the HAR v2; (b) mean monthly rainfall and snowmelt averaged over Kyrgyzstan and Tajikistan extracted from the HAR v2; (c) mean monthly landslide occurrences in Kyrgyzstan and Tajikistan from 2004-2018.

## 3.2 Thresholds of atmospheric triggers for landslides in Kyrgyzstan and Tajikistan

Statistics of different properties of LTEs and NLTEs for rainfall, snowmelt, and rainfall+snowmelt are presented in Fig. 5 in the form of empirical cumulative distribution function (eCDF). Rainfall and snowmelt have a high percentage of events with $I_{mean} = 0$, $I_{max} = 0$, and $Q = 0$. This is because, for landslide events that cannot be detected by only rainfall (orange points in Fig. 3), $I_{mean}$, $I_{max}$, and $Q$ of rainfall for these events were all set to zero. The same procedure was conducted for events that cannot be detected by only snowmelt (blue points in Fig. 3). It can be seen in Fig. 5 that LTEs for both entire events and UTL events have stronger $I_{mean}$ and $I_{max}$, as well as larger $Q$ compared to NLTEs. Besides, snowmelt events have much higher $Q$ but lower $I_{mean}$ and $I_{max}$ than rainfall events, indicating that snowmelt events are in general prolonged and not as intense as rainfall events. Overall, the HAR v2 combined with landslide inventories from GLC and GFLD can distinguish LTEs from NLTEs well and has potential in landslide threshold modeling.

We calibrated thresholds of $I_{mean}$, $I_{max}$, and $Q$ using rainfall, snowmelt, and rainfall+snowmelt as predictors. The procedure was conducted for both entire events and UTL events. Predictive performance is better when using the entire period than just using the UTL period (Table 1), which was also concluded by Leonarduzzi et al. (2017). One of the reasons is that by considering a longer period, $I_{mean}$, $I_{max}$, and especially $Q$ of LTEs generally increase, making it easier to distinguish LTEs from NLTEs. This can also be seen from the eCDFs in Fig. 5. In the eCDF space, the threshold defined by maximizing PSS is the point on the x-axis, where the vertical distance between the LTE curve and the NLTE curve is the largest. eCDFs of UTL events are closer to the NLTE curve than eCDFs of the entire events. Therefore, the maximum PSSs of UTL events are smaller (Fig. 5).The better performance by considering the entire period could also indicate that there exists some uncertainty of landslide timing reported in GLC and GFLD. It can be seen from Table 1 that rainfall+snowmelt has the best predictive performance for both entire events and UTL events. The predictive performance indicating by $d$, PSS, and AUC of the three event properties ($I_{mean}$, $I_{max}$, and $Q$) are quite similar, but using $I_{max}$ as a predictor leads to a lower FAR but also a lower HR when compared with $Q$ and $I_{mean}$.

K-fold cross-validation results for entire events and UTL events are presented in Table A2 and Table A3. Cross-validation reduces the sample size and makes the results more sensitive to outliers. The validation results are in line with the conclusions drawn by calibration: (1) among all predictors, rainfall+snowmelt has the best predictive performance for both entire events and UTL events; (2) predictive performance is better when using the entire period than just using the UTL period; (3) predictive performance of $I_{mean}$, $I_{max}$, and $Q$ for rainfall+snowmelt are quite similar, but $I_{max}$ has a lower FAR and also a lower HR.

## 3.3 Mean annual exceedance

Using the thresholds defined in section 3.2 for rainfall+snowmelt UTL events, Figure 6 presents the annual number of rainfall+snowmelt events that exceed the thresholds of $I_{mean} = 5.05\,\mathrm{mm\,d^{-1}}$, $I_{max} = 14.05\,\mathrm{mm\,d^{-1}}$, and $Q = 15.65\,\mathrm{mm}$ (hereafter referred to as $I_{mean,th}$, $I_{max,th}$, and $Q_{th}$ ). Here, only the results for UTL events are presented since the defined thresholds of entire events and UTL events for rainfall+snowmelt are very similar and only deviate within 10%, although their predictive performance is different (Table 1).

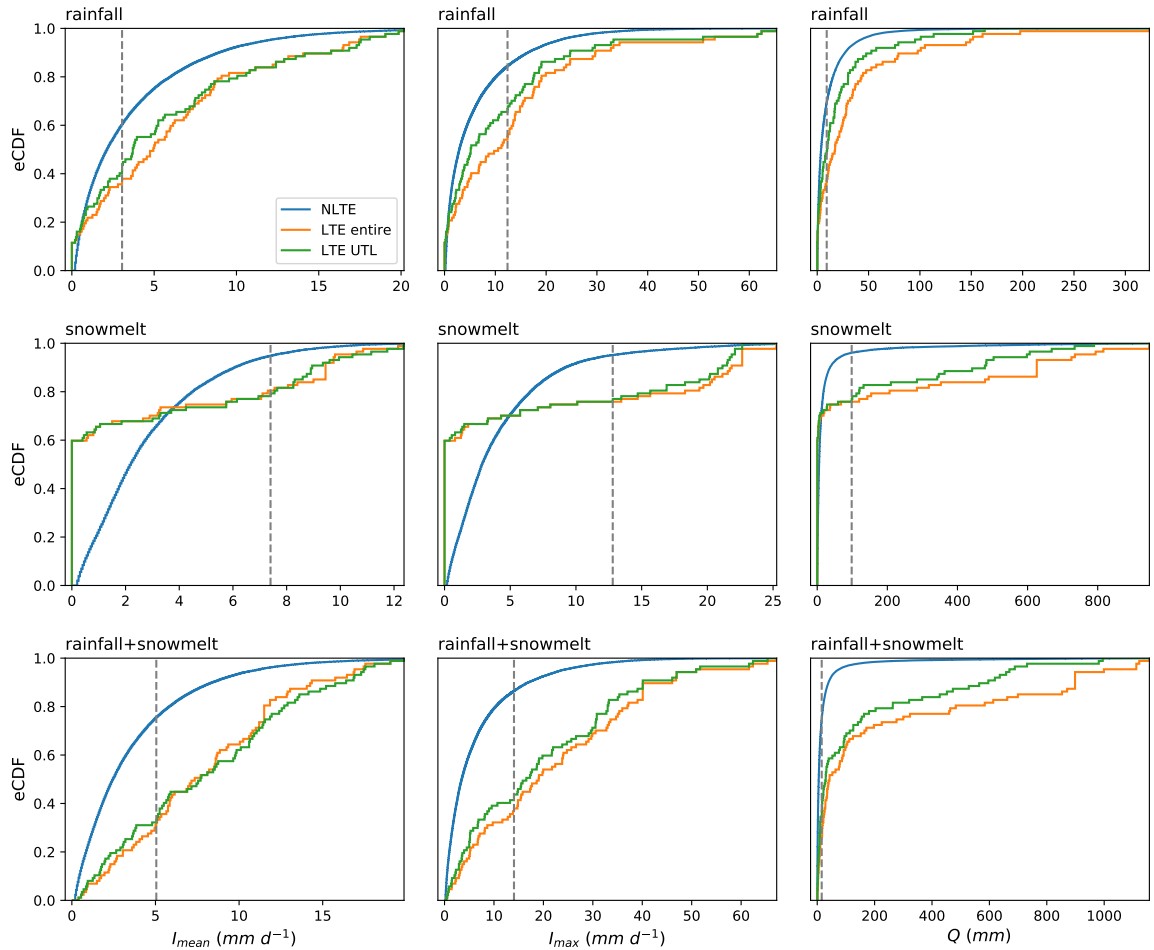

**Figure 5.** eCDF curves of $I_{mean}$, $I_{max}$, $Q$ of NLTE, landslide-triggering entire event (LTE entire), and landslide-triggering up-to-landslide event (LTE UTL) for rainfall, snowmelt, and rainfall+snowmelt during the period of 2004–2018. Grey dashed lines represent the thresholds for UTL event defined in Table 1.

Locations with higher mean annual exceedance over $I_{max,th}$ indicate a higher chance of having rainfall+snowmelt events with high intensity, such as the Fergana Range and the northeastern Tajik Basin. These two regions have a high contribution of rainfall to annual rainfall+snowmelt (Fig. 3), and rainfall events tend to have stronger intensity than snowmelt events (Fig. 5). Locations with high mean annual exceedance over $Q_{th}$ but low exceedance over $I_{max,th}$, including the Pamir Plateau and the Tien Shan, indicate that prolonged events instead of short and intense events are more frequent. The mean annual

exceedance maps of $Q_{th}$ and $I_{mean,th}$ correspond better with the landslide occurrences since they encompass both extreme events and prolonged events. Landslide events reported from the GLC and the GFLD are generally located in areas with high exceedance over $Q_{th}$ and $I_{mean,th}$. However, the mean annual exceedance maps of $Q_{th}$ and $I_{mean,th}$ also have more areas

**Table 1.** Calibrated thresholds of mean intensity $I_{mean}$ $(\mathrm{mm\,d^{-1}})$, maximum intensity $I_{max}$ $(\mathrm{mm\,d^{-1}})$, and accumulated amount $Q$ (mm) for entire events and UTL events of rainfall, snowmelt, and the sum of rainfall and snowmelt (rainfall+snowmelt), and corresponding performance statistics.

| predictor | property | threshold | HR | FAR | $d$ | PSS | AUC |
|---|---|---|---|---|---|---|---|
| rainfall | $I_{mean}$ | 3.60 | 0.62 | 0.35 | 0.51 | 0.27 | 0.62 |
| (entire event) | $I_{max}$ | 11.20 | 0.49 | 0.18 | 0.54 | 0.32 | 0.65 |
|  | $Q$ | 16.95 | 0.52 | 0.18 | 0.52 | 0.34 | 0.67 |
| snowmelt | $I_{mean}$ | 7.05 | 0.23 | 0.06 | 0.77 | 0.17 | 0.31 |
| (entire event | $I_{max}$ | 13.45 | 0.24 | 0.04 | 0.76 | 0.20 | 0.32 |
|  | $Q$ | 119.60 | 0.24 | 0.03 | 0.76 | 0.21 | 0.33 |
| rainfall+snowmelt | $I_{mean}$ | 4.95 | 0.71 | 0.25 | 0.38 | 0.46 | 0.78 |
| (entire event) | $I_{max}$ | 12.80 | 0.67 | 0.15 | 0.37 | 0.51 | 0.81 |
|  | $Q$ | 17.15 | 0.74 | 0.23 | 0.35 | 0.50 | 0.81 |
| rainfall | $I_{mean}$ | 3.05 | 0.60 | 0.40 | 0.57 | 0.20 | 0.59 |
| (UTL event) | $I_{max}$ | 12.40 | 0.34 | 0.16 | 0.67 | 0.19 | 0.58 |
|  | $Q$ | 9.25 | 0.52 | 0.31 | 0.57 | 0.21 | 0.59 |
| snowmelt | $I_{mean}$ | 7.40 | 0.22 | 0.05 | 0.78 | 0.17 | 0.31 |
| (UTL event) | $I_{max}$ | 12.80 | 0.24 | 0.05 | 0.76 | 0.19 | 0.32 |
|  | $Q$ | 98.30 | 0.24 | 0.04 | 0.76 | 0.20 | 0.32 |
| rainfall+snowmelt | $I_{mean}$ | 5.05 | 0.68 | 0.25 | 0.41 | 0.43 | 0.76 |
| (UTL event) | $I_{max}$ | 14.05 | 0.59 | 0.14 | 0.44 | 0.45 | 0.77 |
|  | $Q$ | 15.65 | 0.66 | 0.25 | 0.43 | 0.40 | 0.76 |

with false alarms, i.e., areas with high mean annual exceedance but no landslide occurrence. In remote areas, such as the Tien Shan, high false alarms could be due to the fact that landslides extracted from media reports are generally under-reported in remote regions. This is discussed in detail in section 4.1. In contrast, the mean annual exceedance map of $I_{max,th}$ misses more landslide events but has less false alarm area when compared to the exceedance maps of $Q_{th}$ and $I_{mean,th}$.

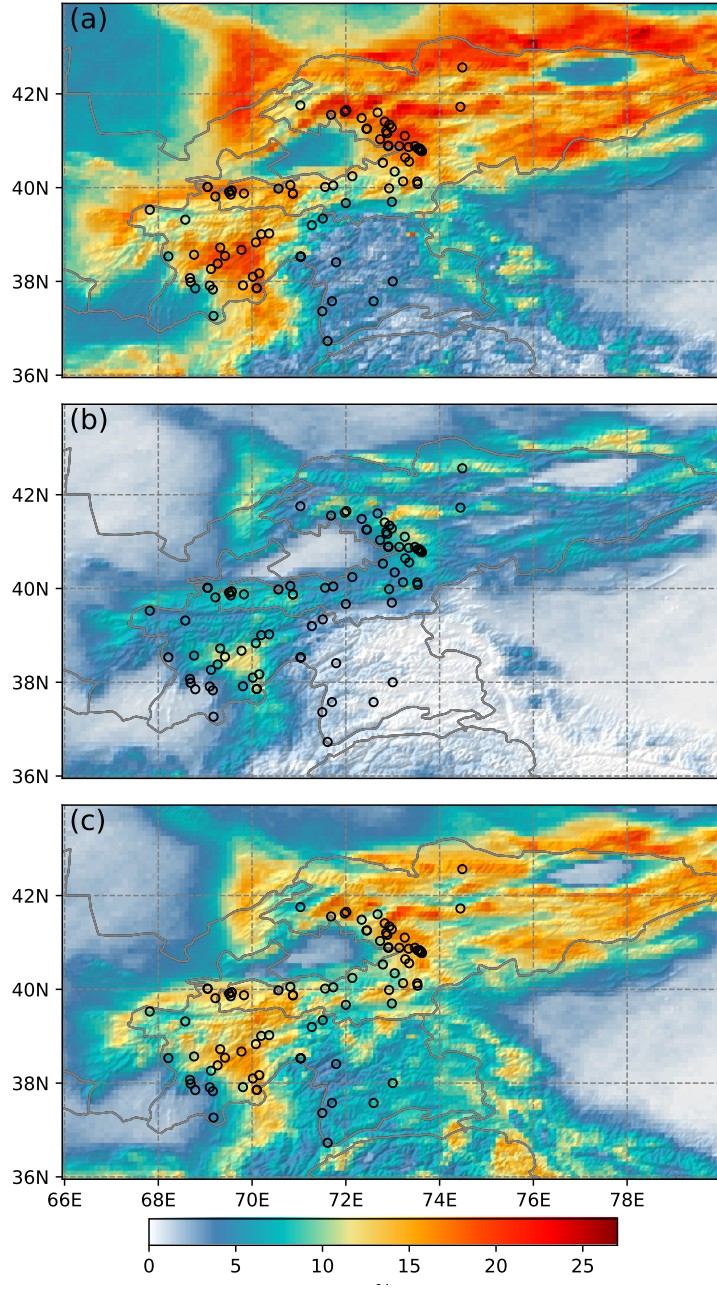

**Figure 6.** Mean annual exceedance (number of events per year) of (a) $I_{mean} = 5.05\,\mathrm{mm\,d^{-1}}$ (b) $I_{max} = 14.05\,\mathrm{mm\,d^{-1}}$; and (c) $Q = 15.65\,\mathrm{mm}$ for the rainfll+snowmelt UTL events. Black circles: landslide events from GLC and GFLD. Topographic shading is based on DEM data from SRTM.

## 4 Discussion

### 4.1 Sources of uncertainty

The uncertainty of the results depends on the accuracy of the data and the method applied to analyze the data. Our approach is purely empirical-based, which allows us to investigate broader areas without knowing the detailed surface characteristics of each landslide event. However, slope instability often results from numerous factors. The interaction between non-climatic characteristics and atmospheric triggers is also responsible for the initiation of landslides (Berti et al., 2012; Jia et al., 2020), which can not be captured by empirical methods. This is the reason why not all rainfall+snowmelt events that exceed $I_{mean,th}$, $I_{max,th}$, and $Q_{th}$ triggered landslides (Fig. 6), even though the number of landslides is underestimated.

Uncertainty in landslide inventories and atmospheric data is a very common issue in studies investigating thresholds for landslide triggering. These two sources of uncertainty have been comprehensively discussed and quantified (e.g., Nikolopoulos et al., 2014, 2015; Marra et al., 2016, 2017; Rossi et al., 2017; Peres et al., 2018; Marra, 2019). Uncertainty in these two data sources generally results in an underestimation of rainfall thresholds, leading to a higher false alarm rate (Nikolopoulos et al., 2014, 2015; Marra et al., 2016; Peres et al., 2018). In the following subsections, we discuss the uncertainty stemming from the landslide inventories (GLC and GFLD) and the rainfall and snowmelt in the HAR v2.

### 4.1.1 Uncertainty of landslide inventories

Uncertainties of the GLC and GFLD are comprehensively discussed in Kirschbaum et al. (2010), Kirschbaum et al. (2015), and Froude and Petley (2018). The first major problem of these two data sets is that they underestimate the total number of landslides. This is because these two data sets' primary sources are media reports, which are biased towards events with human casualties (Carrara et al., 2003). The second issue is that the spatial distribution of landslides is biased towards populated areas. In our study area, landslide events also tend to cluster in areas with high population density, e.g., the eastern rim of the Fergana Basin and the Tajik Basin. Landslide number over remote areas is much likely to be under-reported. In addition, there is large uncertainty in landslide location because most media reports do not contain the exact location where landslides were initiated, but rather just the name of the village, road, or city affected by landslides. An example in our case is the landslide event in the Issyk-Kul Basin (Fig. 1), the location of which is in a flat area, and the location accuracy provided by the GLC is "exact". This landslide event's initial zone must be different from the reported location and somewhere nearby with slopes. We also failed to determine the climatic trigger of this landslide event using the HAR v2. Last but not least, landslide timing was also reported with a certain degree of uncertainty. Although it is more typical that a landslide was reported after its actual occurrence (positive errors), negative errors are also possible depending on the interpretation of historical landslide information from an analyst (Peres et al., 2018). Our results show that using the entire weather event period leads to a better predictive performance than just using the UTL period (Table 1). This could be an indication of negative errors in the landslide timing.

Despite these known limitations, the GLC and the GFLD still provide the lower boundary of landslide number and are proven to be valuable in global and regional landslide studies. For example, the GLC has been successfully applied to detect the initiation of rainfall-induced landslide globally (Jia et al., 2020), to investigate the spatiotemporal distribution of potential

landslide triggering factors (Stanley et al., 2020), to explore the synoptic-scale precursors of landslides (Hunt and Dimri, 2021), and to evaluate the Global Landslide Hazard Assessment Model (Kirschbaum and Stanley, 2018). Although the landslide number is known to be incomplete, our results show that they can still present the seasonal distribution of landslide occurrence reasonably well (Fig. 4). This was also concluded by Kirschbaum et al. (2015), who stated that the reason for the unbiased seasonal distribution of landslide occurrence is that the compilation method depends on media alerts, which are consistent throughout the year. Additionally, even though location uncertainty exists, we could determine atmospheric triggers of 91% of landslide events (87 out of 96). The reason could be that landslide-triggering rainfall and snowmelt events generally have a large spatial extend (Leonarduzzi et al., 2017).

### 4.1.2 Uncertainty of atmospheric data

Extracting weather data that can represent the exact weather condition at landslide sites is always a challenge in studies investigating rainfall thresholds for landslide triggering. Rain gauges are the main source of rainfall information (Segoni et al., 2018), and it is very seldom that landslide initial locations are gauged. Due to the highly heterogeneous spatial distribution of precipitation, especially over complex terrains, there exists great uncertainty when rainfall is not directly measured from landslide initial points. Additionally, Marra et al. (2016) found that the initial points of shallow landslides and debris flows generally correspond to the local peak of rainfall. Rain depth decreases with the distance, causing an underestimation when rainfall is measured away from the landslide initial point. Traditionally, the nearest gauge is used to represent the weather condition at the landslide site, which sometimes can be kilometers away. Nikolopoulos et al. (2015) examined other more complicated interpolation methods, such as inverse distance weighting and ordinary kriging, and concluded that these methods did not bring any particular added value to the simplest nearest neighbor method.

Using gridded data can avoid this allocation problem (Leonarduzzi et al., 2017). But uncertainties still exist since gridded data only represent the grid-mean value but not the "true" weather condition at landslide sites. Nevertheless, it is still essential that the gridded data used in our study can accurately represent the grid-mean value. The WRF model configurations of the HAR v2, such as the forcing strategy, physical parameterization schemes, were carefully chosen to ensure its quality (Wang et al., 2021). Several studies (Pritchard et al., 2019; Li et al., 2020) indicate the high accuracy and quality of the old version of the High Asia Refined Analysis (HAR) (Maussion et al., 2014). Wang et al. (2021) compared the performance of the two versions of the HAR against in-situ observations from 57 GSOD stations over the High Mountain Asia in terms of daily precipitation and air temperature at $2\,\mathrm{m}$. It was concluded that compared to the old version, HAR v2 generally produces slightly higher precipitation amounts with a mean bias of $0.36\,\mathrm{mm\,d^{-1}}$. Furthermore, Hamm et al. (2020) compared the HAR v2 with other gridded precipitation data sets in different spatial resolutions, including reanalysis data and satellite-based precipitation retrieval, over a rugged terrain of the central Himalaya and the southwestern Tibetan Plateau. It was concluded that the HAR v2 is the only product that can resolve orographic precipitation, which is a fundamental process over complex terrain. Simulation of air temperature at $2\,\mathrm{m}$ in the HAR v2 is better than the old version due to the snow depth correction approach (Wang et al., 2021). Snowmelt in the HAR v2 is resolved by the Noah LSM, which only considers a single layer of snowpack (Koren et al., 1999). Several studies found uncertainty of the Noah LSM in reproducing the snow-related process, e.g., the overestimation of

snow albedo (e.g., Chen et al., 2014; Minder et al., 2016; Tomasi et al., 2017). Nevertheless, the snow-related process is the major weakness of LSMs and needs further improvement in the future (Chen et al., 2014).

### 4.1.3 Impact of spatial resolution of atmospheric data

Previous studies have shown that the spatial resolutions of gridded rainfall data have impacts on identifying landslide triggering thresholds (Marra et al., 2017; Nikolopoulos et al., 2017). To investigate the influence of spatial resolution of rainfall+snowmelt data on the event properties of landslide triggering weather events and the triggering thresholds, we resampled the rainfall+snowmelt data from HAR v2 to lower resolutions ($20\,\mathrm{km}$, $30\,\mathrm{km}$, and $40\,\mathrm{km}$). Then, we repeated the procedure described in section 2.2.2 to determine the event properties of LTE UTL events and their associated thresholds. The results are presented in Fig. 7. There are nine "not detected" events when using the original HAR v2 $10\,\mathrm{km}$ data (Fig. 3), which means the rainfall+snowmelt amounts at these landslide grid points are near zero ($\leq 0.2\,\mathrm{mm\,d^{-1}}$) at the day and one day before landslide occurrence. By lowering the spatial resolution, more events can be detected. This implies the uncertainty in the reported landslide location since resampling of rainfall+snowmelt encompasses rainfall+snowmelt information from nearby grid points. In general, $I_{mean}$ and $I_{max}$ decrease with the increase of grid size, which is in line with the findings of Hamm et al. (2020) that higher resolved products generally capture more extreme events than coarser products. $I_{mean}$ and $I_{max}$ thresholds defined by coarser products are also generally lower. The impact of grid size on $Q$ is the opposite: larger grid size leads to higher $Q$ and threshold value. This is closely associated with the increase of event duration with the increase of grid spacing, resulting from the fact that the resampling process can blend several localized events temporally together. However, lowering the spatial resolution does not lead to worse predictive performance. This, on the one hand, implies again that lower resolution can partly compensate for the uncertainty in landslide locations. On the other hand, it indicates that although landslide initiation itself is a highly localized phenomenon, the weather processes that ensure sufficient water input into the system and trigger landslides can be clearly identified at the meso-scale (Prenner et al., 2018).

Based on the above analysis, it can be expected that a convection-permitting scale ($<10\,\mathrm{km}$) downscaling simulation would provide a more realistic representation of weather events that initialized landslides. Compared to such a high-resolution simulation, the HAR v2 $10\,\mathrm{km}$ data would underestimate the intensity and overestimate the duration of landslide triggering rainfall+snowmelt events. Moreover, the $10\,\mathrm{km}$ resolution of the HAR v2 is not able to explicitly resolve convection processes. Convection-permitting scale simulations show improvement over simulations applying cumulus parameterization schemes in several aspects, such as more accurate reproduction of the timing of precipitation peaks (Ou et al., 2020; Zhou et al., 2021). However, a finer resolution has a lower tolerance to the uncertainty in the landslide location. The potential of a kilometer-scale simulation cannot be realized if the landslide location uncertainty is larger than the gird size. Thus, for our study region, future studies should not only focus on acquiring high-resolution and high-quality atmospheric data, but also on developing landslide inventories with higher location accuracy.

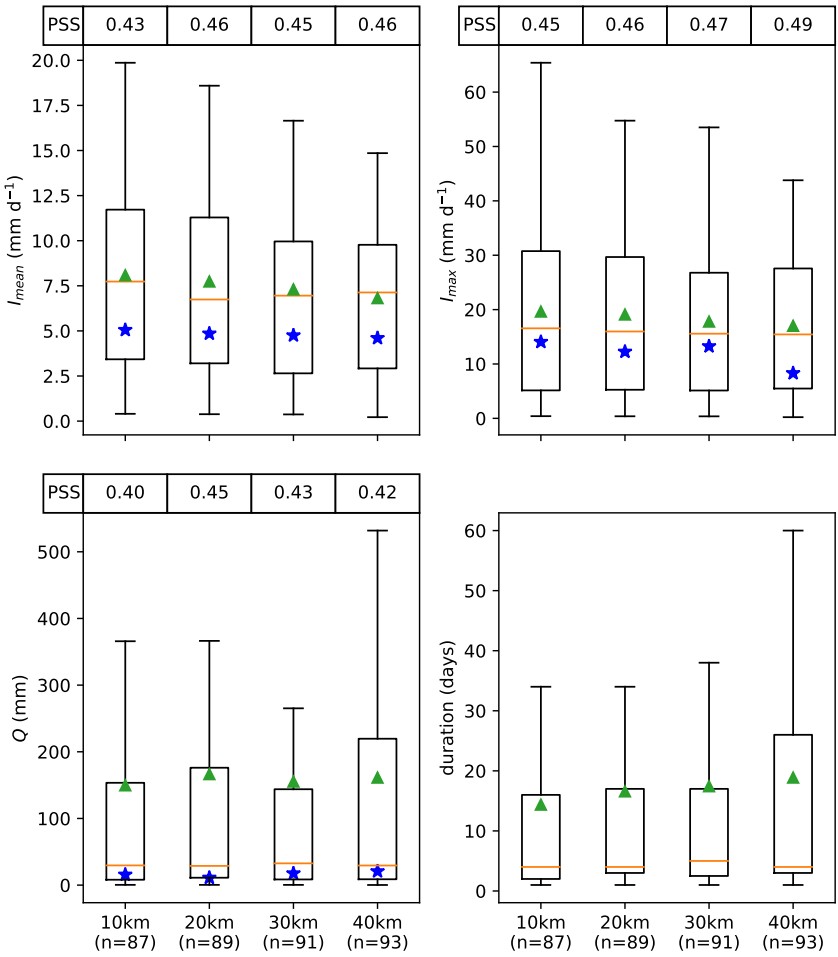

**Figure 7.** Boxplots demonstrating the impact of spatial resolution of atmospheric data on $I_{mean}$, $I_{max}$, $Q$, and duration of LTE UTL events, as well as the associated landslide triggering thresholds (blue stars). The yellow line denotes the median and the green triangle indicates the mean. Outliers are not shown for a better intercomparison. $n$ denotes the number of landslide events detected by rainfall+snowmelt.

## 4.2 Climatic disposition

In probabilistic risk analysis (e.g., Scherer et al., 2013), the risk that a system experiences an adverse effect caused by a hazardous process is given as the product of hazard and vulnerability. Vulnerability itself depends on exposure and sensitivity. Adverse effects only occur when the elements at risk are exposed to a hazardous event. Thus, risk is a function of hazard, exposure, and sensitivity. Applying this risk concept to our case, the adverse effect is landslide triggered by rainfall+snowmelt, and the hazardous process is rainfall+snowmelt events that exceed the defined thresholds. The risk that a location experiences landslide triggered by rainfall+snowmelt depends on two factors: (a) how frequent a location is exposed to rainfall+snowmelt events that exceed $I_{mean,th}$, $I_{max,th}$, and $Q_{th}$, and (b) how sensitive slope instability can be triggered at this location. Climatic

disposition represented by mean annual exceedance is actually factor (a) and comprises both aspects of hazard and exposure. Sensitivity is non-climatic landslide susceptibility that is only controlled by terrestrial characteristics. Thus, to assess landslide
susceptibility, both climatic and non-climatic aspects need to be included.

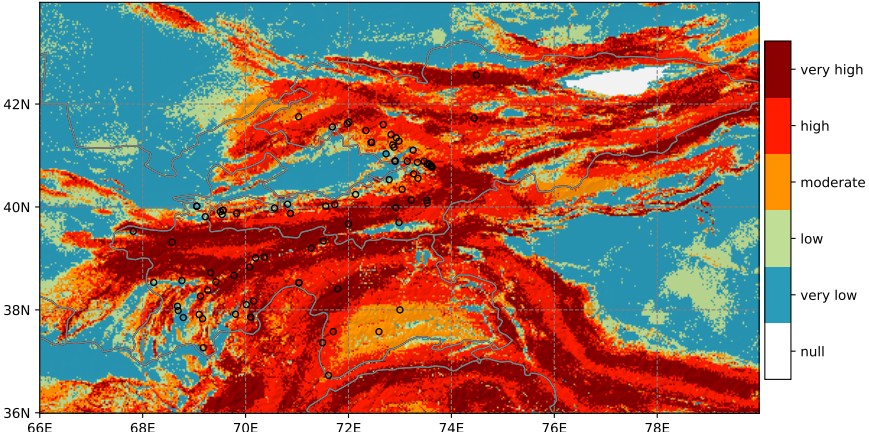

**Figure 8.** Non-climatic landslide susceptibility map computed using slope, geology, fault zones, road networks, and forest loss developed by Stanley and Kirschbaum (2017). Black circles: landslide events from GLC and GFLD. Topographic shading is based on DEM data from SRTM.

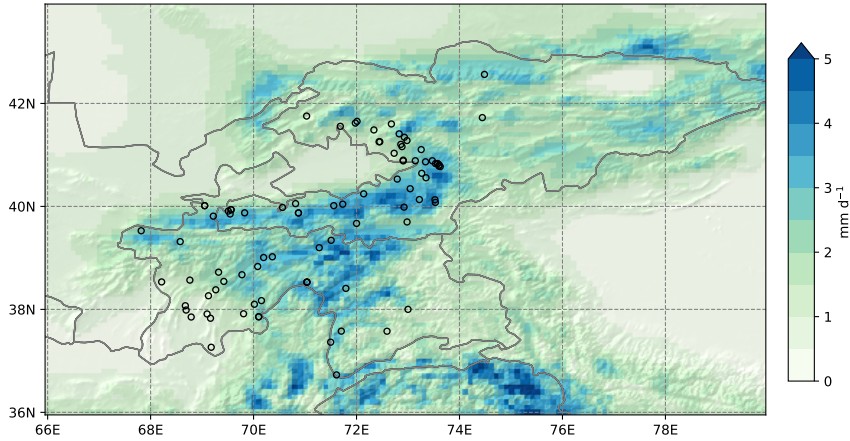

**Figure 9.** Annual sum of rainfall and snowmelt averaged over 2014-2018 from HAR v2. Black circles: landslide events from GLC and GFLD. Topographic shading is based on Digital Elevation Model data from SRTM.

The majority of landslide susceptibility studies only considered non-climatic factors. We compared our mean annul exceedance maps with a non-climatic landslide susceptibility map developed by Stanley and Kirschbaum (2017) at a resolution

of approximately 1 km (Fig. 8). This non-climatic susceptibility map was generated using a heuristic fuzzy approach, in which slope, faults, geology, forest loss, and road networks were taken into account. This map is chosen because it covers the whole

of Kyrgyzstan and Tajikistan. Even though the non-climatic susceptibility map and our mean annual exceedance maps were generated by totally different methods, they share some similarities. They both show higher values over areas with steep slopes and lower values in intermontane basins and valleys. This is because topographic relief is considered the best first-order rainfall predictor (Bookhagen and Strecker, 2008). The non-climatic susceptibility map includes information on topography, and topography is explicitly resolved during dynamical downscaling. Mean annual exceedance maps not only display these local-

scale features caused by topography but also comprise general atmospheric circulation processes. Around 23% of landslide events are located in zones with low and very low susceptibility. Landslide locations with low susceptibility in the eastern and southern rims of the Fergana Basin exhibit high climatic disposition (Fig. 6). This discrepancy between the non-climatic landslide susceptibility and our mean annual exceedance maps suggests that both climatic and non-climatic aspects need to be considered for landslide susceptibility mapping. Some event locations show both low susceptibility and low climatic disposition

(e.g., in southwestern Tajikistan), which implies the uncertainty in reported landslide locations.

In addition, some landslide susceptibility studies took climate into account, but they often simply applied averaged annual precipitation (e.g., Shahabi et al., 2014; Havenith et al., 2015b; Wang et al., 2015). Averaged annual precipitation only shows the climatological conditions in general. Mean annual exceedance is derived from weather-scale triggering conditions, and therefore, it also contains information on extreme processes. In our case, for instance, the mean annual rainfall+snowmelt map

does not correspond well with landslide occurrences, especially in the Tajik Basin and the northeastern rim of the Fergana Basin (Fig. 9). But these landslide events are captured better in both mean annual exceedance maps (Fig. 6). This indicates the added value of climatic disposition derived from triggering conditions.

### 4.3 Thresholds for different landslide size

GLC provides six categorized landslide sizes. Landslide events in Kyrgyzstan and Tajikistan fall into the following categories:

(1) small: small landslide affecting one hill slope or small area; (2) medium: moderately sized landslide that could be either a single event or multiple landslides within an area, and involves a large volume of material;(3) large: large landslide or series of landslides that occur in one general area but cover a wide area; (4) unknown (Kirschbaum et al., 2015). GFLD does not contain information about landslide size. Therefore, for landslide events from GFLD, we set the landslide size as "unknown". Table 2 presents the calibrated thresholds and corresponding statistical scores for these categories for UTL events. Using entire events

leads to similar results (not presented here).

Interestingly, the thresholds for landslides with small sizes are higher than other categories and have the best predictive performance. All these 5 small-sized landslide events are snowmelt contributed events that occurred from March to May. The worse predictive performance for landslides with larger sizes could indicate that for those events, the triggering mechanism is much more complicated than small-sized events, and other non-atmospheric factors might also play a role. However, the

370 sample size of small-sized landslide events is too small to draw a robust conclusion. The number of small-sized landslides is expected to be under-reported since media reports are biased towards events with more severe impacts.

**Table 2.** Calibrated thresholds of $I_{mean}$ ($\mathrm{mm\,d^{-1}}$), $I_{max}$ ($\mathrm{mm\,d^{-1}}$), and $Q$ (mm) for UTL events of the sum of rainfall and snowmelt (rainfall+snowmelt), and corresponding performance statistics for different categories of landslide size. n refers to the number of landslides in each category.

| landslide size | property | threshold | HR | FAR | $d$ | PSS | AUC |
|---|---|---|---|---|---|---|---|
| small | $I_{mean}$ | 9.85 | 1.00 | 0.07 | 0.07 | 0.93 | 0.97 |
| (n=5) | $I_{max}$ | 21.55 | 1.00 | 0.07 | 0.07 | 0.93 | 0.97 |
| | $Q$ | 124.25 | 1.00 | 0.04 | 0.04 | 0.96 | 0.98 |
| medium | $I_{mean}$ | 4.80 | 0.63 | 0.25 | 0.44 | 0.39 | 0.71 |
| (n=41) | $I_{max}$ | 14.05 | 0.49 | 0.12 | 0.53 | 0.37 | 0.73 |
| | $Q$ | 9.65 | 0.73 | 0.35 | 0.44 | 0.38 | 0.72 |
| large | $I_{mean}$ | 8.10 | 0.55 | 0.11 | 0.47 | 0.44 | 0.72 |
| (n=11) | $I_{max}$ | 21.75 | 0.45 | 0.05 | 0.55 | 0.40 | 0.73 |
| | $Q$ | 2.85 | 1.00 | 0.63 | 0.63 | 0.37 | 0.73 |
| unknown | $I_{mean}$ | 5.25 | 0.77 | 0.26 | 0.35 | 0.51 | 0.80 |
| (n=30) | $I_{max}$ | 13.25 | 0.73 | 0.17 | 0.32 | 0.57 | 0.81 |
| | $Q$ | 16.90 | 0.77 | 0.25 | 0.34 | 0.51 | 0.79 |

## 5 Conclusions

In this study, we combined gridded atmospheric data from the HAR v2 with 87 landslide records extracted from the GLC and the GFLD to analyze rainfall and snowmelt conditions that triggered landslides in Kyrgyzstan and Tajikistan. Thresholds for landslide triggering were determined for different event properties for rainfall, snowmelt, and rainfall+snowmelt. Mean annual exceedance maps were generated based on the defined thresholds.

Monthly landslide counts in Kyrgyzstan and Tajikistan correspond well with the monthly distribution of rainfall+snowmelt. An exception is March when soil temperature at the top soil layer (0-0.1m) and air temperature at $2\,\mathrm{m}$ are both below zero. Investigation of the relationship between landslides and soil temperature could be a topic for future studies. Snowmelt plays a crucial role in landslide triggering in Kyrgyzstan and Tajikistan since it contributes to the triggering of 40% of landslide events.

By including snowmelt as an additional trigger, the skill of landslide prediction was significantly improved. $I_{mean}$, $I_{max}$, and $Q$ have similar predictive performance. Thresholds of $I_{mean} = 5.05\,\mathrm{mm\,d^{-1}}$, $I_{max} = 14.05\,\mathrm{mm\,d^{-1}}$, and $Q = 15.65\,\mathrm{mm}$ for UTL events were defined for landslide triggering in Kyrgyzstan and Tajikistan. Using the entire period of weather events leads to similar threshold values but better predictive performance. This could indicate uncertainty in landslide timing. Mean annual exceedance maps derived from these thresholds depict climatic disposition and have added value in landslide susceptibility mapping.

The majority of previous studies applied rainfall estimates from in-situ gauges or satellite retrievals. Our study demonstrates the potential of the Regional Climate Model (RCM) in landslide prediction. Dynamical downscaling products generated by RCMs can provide physically consistent, high-resolution data that is extremely valuable for data scare areas. Given the global

applicability of the dynamical downscaling method, our approach can also be applied in other regions, as long as the number and quality of landslide records are sufficient. Even though a higher-resolved downscaling product can reproduce landslide-triggering weather events more realistically, it has a lower tolerance to the uncertainty in landslide locations and does not necessarily lead to better predictive performance. Future studies in Kyrgyzstan and Tajikistan should focus on developing landslide inventories with both high location accuracy and timing accuracy to reduce the uncertainty in triggering thresholds.

**Appendix A**

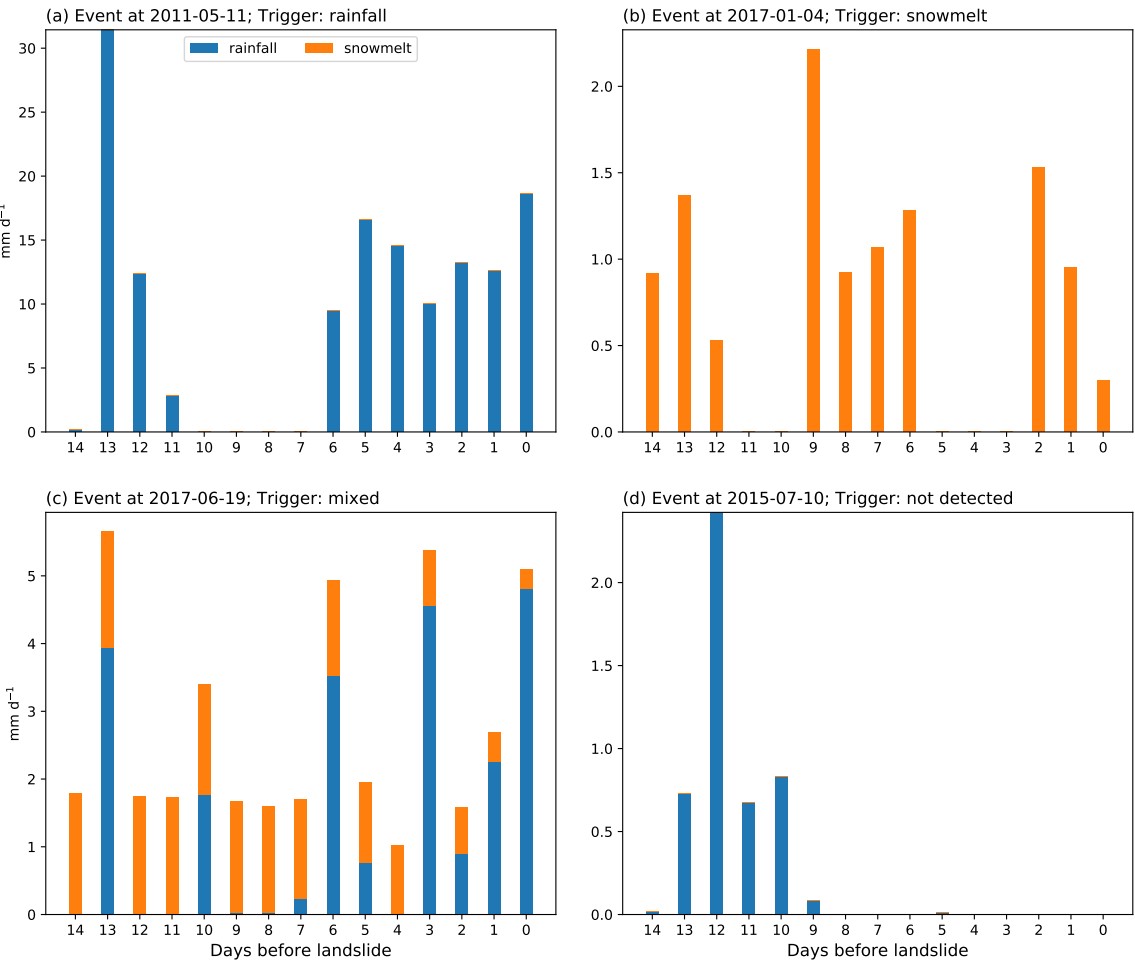

**Figure A1.** Event-based temporal process of rainfall and snowmelt for selected landslide events with landslide triggers defined as (a) "rainfall", (b) "snowmelt"; (c) "mixed"; and (d) "not detected" according to the method described in section 2.2.1.

Table A1: Landslide events in Kyrgyzstan and Tajikistan extracted from GLC and GFLD from 2004 to 2018. Column "trigger" indicates the trigger of landslide events detected by the HAR v2.

| Event date | Source | Longitude | Latitude | Country | Trigger |
|---|---|---|---|---|---|
| 2004-04-17 | GFLD | 73.0420 | 40.3428 | Kyrgyzstan | mixed |
| 2004-05-22 | GFLD | 69.2172 | 39.8106 | Tajikistan | rainfall |
| 2004-06-14 | GFLD | 70.8718 | 39.8734 | Kyrgyzstan | rainfall |
| 2004-11-17 | GFLD | 70.0802 | 38.8324 | Tajikistan | mixed |
| 2005-03-13 | GFLD | 69.0502 | 40.0141 | Tajikistan | mixed |
| 2005-04-09 | GFLD | 69.2656 | 38.3801 | Tajikistan | mixed |
| 2007-03-25 | GLC | 70.1951 | 39.0071 | Tajikistan | mixed |
| 2007-04-01 | GLC | 72.5920 | 37.5760 | Tajikistan | mixed |
| 2007-04-05 | GLC | 71.6110 | 36.7270 | Tajikistan | snowmelt |
| 2007-04-17 | GLC | 71.6849 | 41.5552 | Kyrgyzstan | rainfall |
| 2007-04-17 | GLC | 68.2140 | 38.5330 | Tajikistan | rainfall |
| 2007-04-22 | GLC | 73.1416 | 40.8870 | Kyrgyzstan | rainfall |
| 2007-06-05 | GFLD | 69.1633 | 37.8276 | Tajikistan | rainfall |
| 2007-07-21 | GLC | 73.0000 | 38.0000 | Tajikistan | mixed |
| 2007-07-22 | GLC | 70.4400 | 40.7500 | Tajikistan | not detected |
| 2007-07-22 | GFLD | 71.0363 | 38.5289 | Tajikistan | rainfall |
| 2009-04-16 | GFLD | 71.9767 | 41.6184 | Kyrgyzstan | rainfall |
| 2009-04-21 | GLC | 68.7882 | 37.8515 | Tajikistan | rainfall |
| 2009-05-05 | GFLD | 70.1529 | 38.1701 | Tajikistan | rainfall |
| 2009-05-07 | GFLD | 69.7741 | 38.6726 | Tajikistan | rainfall |
| 2009-05-11 | GFLD | 71.0363 | 38.5289 | Tajikistan | snowmelt |
| 2009-05-14 | GLC | 68.6900 | 37.9867 | Tajikistan | rainfall |
| 2009-05-16 | GFLD | 71.0363 | 38.5289 | Tajikistan | snowmelt |
| 2009-05-20 | GFLD | 69.3199 | 38.7221 | Tajikistan | rainfall |
| 2010-03-13 | GFLD | 69.0502 | 40.0141 | Tajikistan | snowmelt |
| 2010-05-07 | GLC | 69.8054 | 37.9148 | Tajikistan | rainfall |
| 2010-05-07 | GFLD | 70.0994 | 37.8560 | Tajikistan | rainfall |
| 2010-06-03 | GLC | 72.9227 | 39.9854 | Kyrgyzstan | mixed |
| 2011-05-11 | GLC | 72.8282 | 41.4088 | Kyrgyzstan | rainfall |
| 2011-06-12 | GLC | 69.1238 | 38.2644 | Tajikistan | rainfall |
| 2011-06-12 | GLC | 69.5667 | 39.9342 | Kyrgyzstan | rainfall |

| | | | | | |
|---|---|---|---|---|---|
| 2012-05-12 | GLC | 70.8159 | 40.0538 | Kyrgyzstan | rainfall |
| 2012-05-13 | GFLD | 70.8718 | 39.8734 | Kyrgyzstan | rainfall |
| 2013-06-28 | GLC | 72.0106 | 41.6518 | Kyrgyzstan | rainfall |
| 2014-04-12 | GLC | 69.0971 | 37.9107 | Tajikistan | rainfall |
| 2014-04-12 | GFLD | 70.0994 | 37.8560 | Tajikistan | rainfall |
| 2014-04-16 | GFLD | 68.6749 | 38.0710 | Tajikistan | rainfall |
| 2014-04-26 | GFLD | 68.7626 | 38.5685 | Tajikistan | rainfall |
| 2015-04-03 | GFLD | 69.4222 | 38.5428 | Tajikistan | rainfall |
| 2015-05-08 | GLC | 70.0162 | 38.0991 | Tajikistan | rainfall |
| 2015-05-24 | GLC | 72.9053 | 40.8986 | Kyrgyzstan | rainfall |
| 2015-05-24 | GFLD | 73.2559 | 41.1036 | Kyrgyzstan | rainfall |
| 2015-07-10 | GLC | 70.4275 | 39.0712 | Tajikistan | not detected |
| 2015-07-16 | GLC | 71.7041 | 37.5773 | Tajikistan | rainfall |
| 2015-07-21 | GFLD | 71.7929 | 38.4071 | Tajikistan | rainfall |
| 2016-04-26 | GLC | 72.9071 | 40.8894 | Kyrgyzstan | not detected |
| 2016-05-09 | GLC | 68.5748 | 39.3160 | Tajikistan | mixed |
| 2016-05-15 | GLC | 72.9293 | 41.3431 | Kyrgyzstan | rainfall |
| 2016-05-23 | GLC | 72.7907 | 40.5304 | Kyrgyzstan | rainfall |
| 2016-05-27 | GLC | 69.8266 | 39.8751 | Kyrgyzstan | rainfall |
| 2016-05-28 | GLC | 71.5577 | 40.0150 | Kyrgyzstan | mixed |
| 2016-06-16 | GLC | 72.3374 | 41.4850 | Kyrgyzstan | rainfall |
| 2016-06-20 | GLC | 73.5233 | 40.1293 | Kyrgyzstan | rainfall |
| 2016-06-27 | GLC | 74.4438 | 41.7246 | Kyrgyzstan | rainfall |
| 2016-06-29 | GLC | 73.1415 | 41.7649 | Kyrgyzstan | not detected |
| 2016-07-29 | GLC | 69.5597 | 39.9377 | Kyrgyzstan | rainfall |
| 2016-08-16 | GLC | 78.3019 | 42.6831 | Kyrgyzstan | not detected |
| 2016-08-18 | GLC | 70.5626 | 39.9790 | Tajikistan | rainfall |
| 2017-01-04 | GLC | 71.9999 | 39.6699 | Kyrgyzstan | snowmelt |
| 2017-01-26 | GLC | 72.8834 | 40.8960 | Kyrgyzstan | not detected |
| 2017-03-26 | GFLD | 73.5725 | 40.8316 | Kyrgyzstan | mixed |
| 2017-04-07 | GLC | 73.6257 | 40.7733 | Kyrgyzstan | snowmelt |
| 2017-04-09 | GLC | 73.5335 | 40.8320 | Kyrgyzstan | snowmelt |
| 2017-04-10 | GLC | 69.5091 | 39.9095 | Kyrgyzstan | mixed |
| 2017-04-11 | GLC | 72.8601 | 41.2047 | Kyrgyzstan | mixed |
| 2017-04-14 | GFLD | 73.5725 | 40.8316 | Kyrgyzstan | mixed |

| | | | | | |
|---|---|---|---|---|---|
| 2017-04-16 | GLC | 73.2668 | 40.6430 | Kyrgyzstan | snowmelt |
| 2017-04-16 | GLC | 73.6000 | 40.7836 | Kyrgyzstan | snowmelt |
| 2017-04-17 | GLC | 73.6047 | 40.8044 | Kyrgyzstan | mixed |
| 2017-04-18 | GLC | 71.4973 | 37.3628 | Tajikistan | mixed |
| 2017-04-18 | GLC | 72.9069 | 40.8838 | Kyrgyzstan | rainfall |
| 2017-04-22 | GLC | 73.3402 | 40.8663 | Kyrgyzstan | mixed |
| 2017-04-23 | GLC | 71.5074 | 39.3410 | Tajikistan | snowmelt |
| 2017-04-23 | GLC | 72.8835 | 41.1610 | Kyrgyzstan | rainfall |
| 2017-04-23 | GFLD | 72.9801 | 41.2790 | Kyrgyzstan | mixed |
| 2017-04-29 | GLC | 73.4724 | 40.8864 | Kyrgyzstan | mixed |
| 2017-04-29 | GFLD | 73.2203 | 40.1325 | Kyrgyzstan | mixed |
| 2017-04-30 | GLC | 72.4381 | 41.2550 | Kyrgyzstan | rainfall |
| 2017-04-30 | GLC | 73.5310 | 40.0774 | Kyrgyzstan | mixed |
| 2017-05-10 | GLC | 74.4847 | 42.5635 | Kyrgyzstan | mixed |
| 2017-05-11 | GLC | 73.3497 | 40.5560 | Kyrgyzstan | rainfall |
| 2017-05-16 | GLC | 71.0302 | 41.7545 | Kyrgyzstan | rainfall |
| 2017-05-17 | GLC | 72.6771 | 41.6014 | Kyrgyzstan | rainfall |
| 2017-05-28 | GLC | 71.2755 | 39.1978 | Tajikistan | mixed |
| 2017-06-19 | GLC | 72.9814 | 39.6978 | Kyrgyzstan | mixed |
| 2017-06-19 | GLC | 71.7318 | 40.0439 | Kyrgyzstan | rainfall |
| 2017-06-26 | GLC | 67.8173 | 39.5267 | Tajikistan | rainfall |
| 2017-06-28 | GLC | 68.5480 | 39.3951 | Tajikistan | not detected |
| 2017-06-29 | GLC | 72.7303 | 41.0321 | Kyrgyzstan | rainfall |
| 2017-06-29 | GLC | 72.4521 | 41.2557 | Kyrgyzstan | rainfall |
| 2017-07-03 | GLC | 70.3650 | 39.0219 | Tajikistan | rainfall |
| 2017-07-03 | GLC | 68.4838 | 39.1172 | Tajikistan | not detected |
| 2017-07-04 | GLC | 69.5279 | 39.8102 | Kyrgyzstan | not detected |
| 2018-05-13 | GLC | 69.5445 | 39.8526 | Kyrgyzstan | rainfall |
| 2018-05-16 | GLC | 69.1773 | 37.2642 | Tajikistan | rainfall |
| 2018-05-21 | GLC | 72.1386 | 40.2437 | Kyrgyzstan | mixed |

**Table A2.** K-fold validation results. Mean values and standard deviations (in parentheses) for thresholds of $I_{mean}$ $(\mathrm{mm\,d^{-1}})$, $I_{max}$ $(\mathrm{mm\,d^{-1}})$, and $Q$ (mm) for entire events of rainfall, snowmelt, and the sum of rainfall and snowmelt (rainfall+snowmelt), and corresponding performance statistics.

| predictor | property | threshold | HR | FAR | $d$ | PSS | AUC |
|---|---|---|---|---|---|---|---|
| rainfall | $I_{mean}$ | 3.76 | 0.56 | 0.33 | 0.56 | 0.23 | 0.62 |
| | | (0.33) | (0.14) | (0.03) | (0.10) | (0.13) | (0.01) |
| | $I_{max}$ | 11.06 | 0.46 | 0.18 | 0.57 | 0.28 | 0.65 |
| | | (0.66) | (0.16) | (0.02) | (0.15) | (0.15) | (0.01) |
| | $Q$ | 12.31 | 0.53 | 0.25 | 0.55 | 0.27 | 0.67 |
| | | (3.88) | (0.16) | (0.07) | (0.10) | (0.10) | (0.01) |
| snowmelt | $I_{mean}$ | 7.06 | 0.22 | 0.06 | 0.78 | 0.16 | 0.31 |
| | | (0.02) | (0.14) | (0.01) | (0.14) | (0.14) | (0.02) |
| | $I_{max}$ | 13.61 | 0.23 | 0.04 | 0.77 | 0.19 | 0.32 |
| | | (0.44) | (0.13) | (0.01) | (0.13) | (0.12) | (0.01) |
| | $Q$ | 122.38 | 0.23 | 0.03 | 0.77 | 0.20 | 0.33 |
| | | (7.93) | (0.13) | (0.01) | (0.13) | (0.12) | (0.01) |
| rainfall+snowmelt | $I_{mean}$ | 4.96 | 0.70 | 0.25 | 0.40 | 0.45 | 0.78 |
| | | (0.02) | (0.13) | (0.02) | (0.08) | (0.14) | (0.01) |
| | $I_{max}$ | 12.93 | 0.65 | 0.15 | 0.39 | 0.49 | 0.81 |
| | | (0.37) | (0.15) | (0.01) | (0.13) | (0.15) | (0.01) |
| | $Q$ | 17.20 | 0.71 | 0.23 | 0.38 | 0.48 | 0.81 |
| | | (0.14) | (0.15) | (0.02) | (0.10) | (0.13) | (0.01) |

**Table A3.** K-fold validation results. Mean values and standard deviations (in parentheses) for thresholds of $I_{mean}$ $(\mathrm{mm\,d^{-1}})$, $I_{max}$ $(\mathrm{mm\,d^{-1}})$, and $Q$ (mm) for UTL events of rainfall, snowmelt, and the sum of rainfall and snowmelt (rainfall+snowmelt), and corresponding performance statistics.

| predictor | property | threshold | HR | FAR | $d$ | PSS | AUC |
|---|---|---|---|---|---|---|---|
| rainfall | $I_{mean}$ | 4.04 | 0.45 | 0.33 | 0.66 | 0.12 | 0.59 |
| | | (1.47) | (0.13) | (0.10) | (0.08) | (0.08) | (0.01) |
| | $I_{max}$ | 10.94 | 0.34 | 0.18 | 0.68 | 0.16 | 0.58 |
| | | (1.47) | (0.06) | (0.04) | (0.05) | (0.06) | (0.01) |
| | $Q$ | 10.21 | 0.46 | 0.29 | 0.62 | 0.17 | 0.59 |
| | | (2.22) | (0.09) | (0.04) | (0.09) | (0.11) | (0.01) |
| snowmelt | $I_{mean}$ | 7.14 | 0.21 | 0.06 | 0.79 | 0.15 | 0.31 |
| | | (0.26) | (0.10) | (0.02) | (0.10) | (0.09) | (0.02) |
| | $I_{max}$ | 12.88 | 0.23 | 0.05 | 0.77 | 0.18 | 0.32 |
| | | (0.23) | (0.12) | (0.01) | (0.12) | (0.11) | (0.02) |
| | $Q$ | 99.95 | 0.22 | 0.04 | 0.78 | 0.18 | 0.32 |
| | | (4.67) | (0.13) | (0.01) | (0.13) | (0.13) | (0.02) |
| rainfall+snowmelt | $I_{mean}$ | 5.35 | 0.61 | 0.23 | 0.47 | 0.38 | 0.76 |
| | | (0.85) | (0.22) | (0.04) | (0.17) | (0.18) | (0.01) |
| | $I_{max}$ | 13.54 | 0.56 | 0.14 | 0.47 | 0.42 | 0.77 |
| | | (0.56) | (0.15) | (0.01) | (0.14) | (0.14) | (0.01) |
| | $Q$ | 15.83 | 0.63 | 0.25 | 0.45 | 0.38 | 0.76 |
| | | (0.44) | (0.13) | (0.02) | (0.10) | (0.12) | (0.01) |

*Code and data availability.* The landslide data and atmospheric data used in this study are freely available from the following links:

– Global Landslide Catalog (GLC): https://maps.nccs.nasa.gov/arcgis/home/item.html?id=eec7aee8d2e040c7b8d3ee5fd0e0d7b9

– Global Fatal Landslide Database (GFLD): https://blogs.agu.org/landslideblog/2019/06/18/global-fatal-landslide-database-1/

– High Asia Refined Analysis version 2 (HAR v2): https://www.klima.tu-berlin.de/HARv2

The source code used in this study is freely available upon request.

*Author contributions.* All authors were involved in study conceptualization and writing of the manuscript. XW collected the data, carried out the analyses, and produced the visualizations.

*Competing interests.* The authors declare that they have no conflict of interest

*Acknowledgements.* This work was supported by the German Federal Ministry of Education and Research (BMBF) under the framework of
405 the "Climatic and Tectonic Natural Hazards in Central Asia (CaTeNA)" project (Grant Number FKZ 03G0878G).

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
