# Peer review of "Atmospheric triggering conditions and climatic disposition of landslides in Kyrgyzstan and Tajikistan at the beginning of the 21st century"

_Natural Hazards and Earth System Sciences, 2020_

## Author Comment (AC1)

**Dear Editor,**

We appreciate your coordination of two helpful and constructive reviews of our manuscript. We are confident that the referees' feedback and comments will help to eliminate imprecision and improve the manuscript. Below, we reply to their comments point by point. The reviewer comments are highlighted in black, while our responses are highlighted in blue. We numbered the reviewer comments to make it easier to refer to a response to another specific comment. R1C1, for example, stands for referee 1, comment 1. Figures and tables used in our response are included at the end of the document. We have already implemented most of the comments in the revised manuscript. We look forward to your editorial consideration of our response.

Best regards, Xun Wang On behalf of all authors

**#1 Referee Comments**

**General comments**

This paper combined gridded atmospheric data from HAR v2 with 87 landslides records from GLC and GFLD to evaluate the critical condition that triggered landslides in Kyrgyzstan and Tajikistan. The results show the significant role of snowmelt in landslide triggering processes. The optimal thresholds of landslide for the sum of rainfall and snowmelt were assessed as well. Overall speaking, this paper is well organized and the research results are valuable for understanding the landslide occurrence caused by mixed triggering factors. However, some issues in my opinion need further clarification prior to it can be considered for publication.

AR: We thank the referee for the review and the constructive comments. We address here all the points raised in the review.

R1C1: The first issue is the limitation of landslide inventory, the landslide record extracted from GLC and GFLD mainly based on media report (more concentrated in the populated area) and mixed-up new landslides and recurrent landslides. This limitation will directly affect the representative of obtained thresholds in the entire study area. In addition, the critical thresholds for different landslide type (new landslide and recurrent landslide) should be discussed separately. Thus more solid and spatial representative landslide inventory that can cover all study area obtained from remote sensing identification is necessary.

AR: GLC and GFLD are, unfortunately, our only choice, since they are the only data sets in the study area that contain the exact date of landslide occurrence. We are fully aware of the limitations of these two data sets and their uncertainties are discussed transparently in L217-235 in the manuscript.

There are indeed other landslide inventories in our study region, such as the Tien Shan Geohazards Database (Havenith et al., 2015a, b) and the multi-temporal landslide inventory from Behling and Roessner (2020). But these two data sets do not have exact dates of landslide occurrence and therefore cannot be applied in the specific purpose of this study. As stated in L31-33 in the manuscript: "Given the highly dynamic nature of weather phenomena, at least a daily time stamp of landslide records is required to investigate weather conditions that trigger landslides."

If the referee is aware of any more solid and spatial representative landslide inventory that contains

exact dates of landslides, we would be very happy to include it in our study.

R1C2: The second issue is how to prove the spatial resolution (10 km) of the gridded atmospheric data from HAR v2 is sufficient to precisely reflect the rainfall or snowmelt condition in each landslide site. AR: Extracting data that can represent the weather condition in landslide sites has always been a challenge. Traditionally, the nearest rain gauge is used, which is sometimes a few kilometers away. Gridded data also do not represent the exact weather condition at landslide sites, but rather the gridmean condition. As also requested by referee #2, we have added a discussion of uncertainties stemming from gridded data. Please see our response to R2C2 for further information.

Among all the available gridded data sets in our study region, we believe the HAR v2 10 km product is the best choice. The accuracy and uncertainty of HAR v2 are discussed in detail in L236-248 in the manuscript. Specifically, "Hamm et al. (2020) compared the HAR v2 with other gridded precipitation data sets, including reanalysis data and satellite-based precipitation retrieval. It was concluded that HAR v2 is the only product that can resolve orographic precipitation, which is a fundamental process over complex terrain. Furthermore, the HAR v2 can capture more extreme precipitation events than coarser product".

Figure 5 in the manuscript shows that despite uncertainties, a grid spacing of 10 km is sufficient to distinguish landslide triggering events from non-landslide triggering events. This indicates that although landslide initiation itself is a highly local phenomenon, the weather processes that ensure sufficient water input into the system and trigger landslides can be clearly identified at a larger scale.

R1C3: The third, the surface runoff, subsurface runoff and ground water caused by rainfall and snowmelt have different time lag characteristic, why these two factors can directly be summed up? Please plot the temporal hydrograph of rainfall and snowmelt respectively, make sure the individual time lag then proceed further calculation.

AR: It is common to directly sum up rainfall and snowmelt to represent the total water input into a system (e.g., Meyer et al., 2012; Bí et al., 2016; Penna et al., 2016; Krøgli et al., 2018; Hammond and Kampf, 2020). The sum of rainfall and snowmelt is also referred to as "water supply" (Meyer et al., 2012; Krøgli et al., 2018) and "total water content" (Bí et al., 2016), and was applied to establish thresholds for landslide triggering (Meyer et al., 2012; Bí et al., 2016; Krøgli et al., 2018).

We agree that rainfall and snowmelt have different time lag characteristics. But investigating the detailed hydrological response to total water input is out of the scope of our study and is also not feasible since our study area is too large. Detailed hydrological conditions, along with other information, such as lithological, morphological, and soil characteristics, are usually considered in studies applying physically based, process-driven models over smaller study areas (Guzzetti et al., 2007; Giannecchini et al, 2016).

R1C4: Except the current thresholds evaluated for the mixed-up landslide occurrence, I suggest the authors further discussing the occurrence thresholds for different landslide types (new landslide and recurrent landslide), and the relation with landslide scale (area) in order to strengthen the contribution of this paper.

AR: Information of whether a landslide is new or recurrent is unfortunately not included in GLC or GFLD. Therefore, we are not able to investigate the thresholds for different landslide types. But we will

discuss the uncertainty in this regard in the revised manuscript.

GLC provides six categorized landslide sizes: unknown, small, medium, large, very large, and catastrophic (Kirschbaum et al., 2015). Landslide events in Kyrgyzstan and Tajikistan fall into the following categories:

- 1) Small: small landslide affecting one hillslope or small area; minimal impacts to infrastructure and roads, no fatalities;
- Medium: moderately sized landslide that could be either a single event or multiple landslides within an area, and involves a large volume of material; moderate impact to infrastructure and roads, may result in fatalities;
- 3) Large: large landslide or series of landslides that occur in one general area but cover a wide area; substantial impacts to infrastructure and roads, likely moderate to high number of fatalities;
- 4) Unknown.

GFLD does not contain information about landslide size. Therefore, for landslide events from GFLD, we set the landslide size as "unknown". Table 1 and Table 2 present the calibrated thresholds and corresponding statistical scores for these categories for entire events and UTL events, respectively. Interestingly, the thresholds for landslides with small sizes are higher than other categories and have the best predictive performance. All these 5 small-sized landslide events are snowmelt contributed events that occurred from March to May. The worse predictive performance for landslides with larger sizes could indicate that for those events, the triggering mechanism is much more complicated than small-sized events, and other non-atmospheric factors might also play a role. However, the sample size of small-sized landslide events is too small to draw a robust conclusion. The number of small-sized landslides is expected to be underreported since media reports are biased towards events with more severe impacts. We have added a new section in the discussion part in the revised manuscript to discuss the relation with landslide sizes.

**Specific comments**

R1C5: P. 5, Ln 117-118, "....landslide event occurred during or one day after a rainfall event...." Why limit to only one day, please give more explanation or references.

AR: In event-based analysis of rainfall threshold, a rainfall event is usually defined as a landslide triggering event if a landslide occurs during or immediately after the rainfall event (e.g., Giannecchini et al., 2016; Leonarduzzi et al., 2017 and 2020). We followed this definition and since the temporal resolution of our atmospheric data is daily, we set the time relaxation to one day.

**R1C6: P. 6, Ln 144, "...events were randomly split into k folds with k=8..." Why select k=8?**

AR: The choice of k is usually 5 or 10 (Kuhn and Johson, 2013). The value for k should be chosen such that (1) each train/test group of data samples is large enough to be statistically representative of the broader dataset; (2) it can separate the dataset as evenly as possible. We have in total 87 landslide events, therefore, k=8 is a good compromise that can fulfill both conditions.

R1C7: P. 7, Ln 161-162, "...Nine landslides events did not occur within any rainfall event, snowmelt event or rainfall + snowmelt event...." This "not detected" uncertainty comes from landslide inventory or observed atmospheric data need more explanation.

AR: The "not detected" events are due to the information mismatch between landslide inventory and the atmospheric data, which stems from the uncertainty in landslide location and timing, as well as the

uncertainty in rainfall and snowmelt simulated in HAR v2. We will add this discussion in section 4.1 "Source of uncertainty".

R1C8: P. 7, Figure 2, what are the subtitle (DJF, MAM, JJA, SON) on the left top means? AR: They stand for seasons as commonly defined in meteorology, spanning three months each: winter (December-February, DJF), spring (March-May, MAM), summer (June-August, JJA), and autumn (September-November, SON). We have added this explanation in section 3.1 after the first sentence and in the caption of Figure 2 in the original manuscript.

R1C9: P. 9, Figure 4, since daily-based atmospheric data were used in this study, and the exact date of landslide occurrence is available. Please select some landslide cases to plot the event-based temporal process of rainfall and snowmelt.

AR: Thanks for the advice! We selected four landslide events with landslide triggers defined as "rainfall", "snowmelt", "mixed" and "not detected" by the method described in section 2.2.1 in the manuscript. We then plotted the temporal process of rainfall and snowmelt from two weeks before the landslide occurrence date (Figure 1). We will include this figure in the revised manuscript in the appendix.

R1C10: P. 10, Ln 184-185, "...Predictive performance is better when using the entire period than just using the UTL period..." Why? The rainfall properties calculated in the period up to the day of landslide occurrence should have more direct contribution to landslide initiation.

AR: The entire period performing better than the UTL period was also concluded in Leonarduzzi et al. (2017). The reasons could be by considering a longer period,  $I_{mean}$ ,  $I_{max}$ , and especially Q of landslide triggering events (LTEs) generally increase, making it easier to distinguish LTEs from non-landslide triggering events (NLTEs). This can be seen from the empirical cumulative distribution function (eCDF, we have changed the name from CDF to eCDF according to R2C4) curves in Figure 5 in the manuscript, the eCDF curves of UTL events are closer to the NLTE curve than eCDFs of the entire events. The better performance by considering the entire period could also indicate that there exists some uncertainty of landslide timing reported in GLC and GFLD. Therefore, in the revised manuscript, we will present the results of both entire events and UTL events in the main text and discuss the reasons behind it transparently. In addition, we will produce climatic disposition maps using thresholds defined by both entire events and UTL events:  $I_{mean}=5.05 \text{ mm d}^{-1}$ ,  $I_{max}=14.05 \text{ mm d}^{-1}$ , Q=15.56 mm), our conclusions regarding climatic disposition stay valid.

**R1C11: P. 16, Figure 7, numbers of historical landslides located in the zone with low to moderate susceptibility. Please give more explanation.**

AR: As stated in L223 - L224 in the manuscript, "there is large uncertainty in landslide location because most media reports do not contain the exact location where landslides initiated, but rather just the name of the village, road, or city affected by landslides". Landslide events located in zones with low susceptibility (e.g., valleys in southwestern Tajikistan, the eastern and southern rims of the Fergana Basin) are very close to slopes with high susceptibility, which could be the actual locations where these

landslide events were initiated.

**Technical corrections**

R1C12: P. 11, Ln 212, "... from median report....." should be "media". AR: Thanks for the comment! We have corrected it.

**#2 Referee Comments**

The paper is well written and structured. There are a couple of things that need to be further specified. AR: We thank the referee for the positive feedback and constructive comments. We address here all the points raised in the review.

R2C1: I would suggest using different terms instead of "climatic disposition/climatic disposition maps". Perhaps "climatic predisposition to landslides/climatic predisposition maps" better explain the concept. AR: The concept "disposition" is commonly used in the context of natural hazards (e.g., Keller and Atzl, 2014; Gill and Malamud, 2014; Theilen-Willige, 2016). The occurrence of natural hazards depends on the disposition and the triggering events. The former refers to "the general setting that favors the specific process", while the latter "leads to threshold crossing of a factor relevant for the hazard incidence" (Malet et al., 2010). Therefore, we think "climatic disposition" is a suitable term to describe the general climatic settings that make slopes prone to failure without actually initiating it. In the revised manuscript, we will explain our concept better in the introduction.

R2C2: Then, it could be useful to describe the limitations in the use of gridded products in detecting atmospheric triggering conditions.

AR: We have added the following discussion in section 4.1:

"Extracting weather data that can represent the exact weather condition at landslide sites is always a challenge in studies investigating rainfall threshold for landslide triggering. Rain gauges are the main source of rainfall information (Segoni et al.,2018), and traditionally, the nearest gauge is used, which sometimes can be kilometers away from the landslide site. Therefore, other more complicated interpolation methods have been developed and their added value has been discussed (Nikolopouloset al., 2015). Using gridded data can avoid this allocation problem (Leonarduzzi et al., 2017). But uncertainties still exist since gridded data only represents the grid-mean value but not the "true" weather condition at landslide sites. Nevertheless, it is still essential that the gridded data used in our study can accurately represent the grid-mean value, especially when landslide locations are reported with some uncertainty."

R2C3: About section 2.1.2 Atmospheric data, I recommend including a more detailed description on how the amount of snowmelt has been calculated using the Surface Energy Balance (SEB) and how the thickness of the existing snow is calculated/taken into account in the snowmelt evaluation.

AR: We are not sure what the reviewer is asking for. In section 2.1.2, we already gave detailed equations of how snowmelt is calculated using variables that can be directly acquired from the HAR v2. These variables are net radiation, sensible heat flux, latent heat flux, ground heat flux, and snow water equivalent. If the reviewer wants to know how these variables are simulated by the Noah Land Surface Model (LSM) in WRF, we will add a brief description of Noah LSM in the revised manuscript.

R2C4: Section 3.2 presents Cumulative distribution function (CDF) curves of mean intensity (Imean),

maximum intensity (Imax), and accumulated amount (Q) of non-landslide triggering event (NLTE), landslide-triggering entire event (LTE entire), and landslide-triggering up-to-landslide event (LTE UTL) for rainfall, snowmelt, and rainfall+snowmelt. What is the probability density function used to calculate the CDF? Please describe.

AR: Fig. 5 actually presents the empirical cumulative distribution function (eCDF), which is based on observations of the data sample. It is not the theoretical cumulative distribution function. We have renamed it to "empirical cumulative distribution function (eCDF)" in the revised manuscript to prevent misunderstandings.

R2C5: Please check line 177-178, the colours of the poins, probably, have been inverted.

AR: The colors of the points are not inverted, but the sentence is not clear and confusing. We wanted to say that orange points in Fig. 3 represent landslide events that cannot be detected by rainfall only, and blue points are landslide events that cannot be detected using snowmelt alone. We have revised this sentence as follows:

"Rainfall and snowmelt have a high percentage of events with  $I_{mean} = 0$ ,  $I_{max} = 0$ , and Q = 0. This is because, for landslide events that cannot be detected by rainfall alone (orange points in Fig. 3),  $I_{mean}$ ,  $I_{max}$ , and Q of rainfall for these events were all set to zero. The same procedure was conducted for events that cannot be detected by snowmelt alone (blue points in Fig. 3)."

R2C6: About figure 5, can you please comment on the fact that for very small values (0,1) of snowmelt for Imax, Imean, Q, there is around 60% probability of having LTE and UTL. It could be useful to exclude from the analysis the rainfall and snowmelt events with Imean = 0, Imax = 0, and Q = 0.

AR: Using only snowmelt as a predictor, only 40% of landslide events (35 out of 87) can be detected. For the remaining 52 events, we set  $I_{mean}$ ,  $I_{max}$ , and Q to zero. Therefore, in Figure 5, there is 60% probability of having LTE and UTL events with  $I_{mean}$ ,  $I_{max}$ , and Q equal to zero. We did not exclude those events with  $I_{mean} = 0$ ,  $I_{max} = 0$ , and Q = 0 because we want to keep the sample size of LTE the same for rainfall, snowmelt, and rainfall+snowmelt such that their skill scores are comparable to each other.

R2C7: About the indicators used to evaluate the thresholds performance, it could be interesting to calculate the efficiency index (TP+TN/TP+FP+FN+TN; please check literature in this regard). It can be seen as a single indicator for quantifying the correct predictions over the total. It could be used in replacement of HR and FAR and compared with the AUC.

AR: Thanks for the suggestion. However, we do not think the efficiency index is a suitable skill score in our case. The efficiency index represents the proportion of correct predictions, which contains two parts: true positive (TP) and true negative (TN). With the increase of threshold values, the value of TN increases, while the value of TP decreases. In our case, take rainfall+snowmelt as an example, there are 55730 non-landslide triggering events (NLTEs), but only 87 landslide triggering events (LTEs). The number of NLTEs takes up more than 99% of the total number of events. Thus, by setting up a large enough threshold, all the NLTEs can be categorized as TN, leading to an efficiency index larger than 99% but a very low value of TP. Due to this huge difference between the numbers of NLTEs and LTEs, we chose the Peirce Skill Score (PSS) as a reference to select thresholds since it is trail-independent, which means it is unbiased even when the numbers of LTEs and NLTEs are not equally presented

(Woodcock, 1976).

R2C8: In section 4. Discussion, could be useful to comment also about the lack of information on the landslide types. Also this aspect increases the uncertainty of the results.

AR: The same concern was also raised by referee #1. Please see our response to R1C4.

**References:**

- Behling, R., & Roessner, S. (2020). Multi-temporal landslide inventory for a study area in Southern Kyrgyzstan derived from RapidEye satellite time series data (2009-2013).
- Bíl, M., Andrášik, R., Zahradníček, P., Kubeček, J., Sedoník, J., & Štěpánek, P. (2016). Total water content thresholds for shallow landslides, Outer Western Carpathians. *Landslides*, *13*(2), 337-347.
- Hamm, A., Arndt, A., Kolbe, C., Wang, X., Thies, B., Boyko, O., ... & Schneider, C. (2020). Intercomparison of Gridded Precipitation Datasets over a Sub-Region of the Central Himalaya and the Southwestern Tibetan Plateau. *Water*, *12*(11), 3271.
- Hammond, J. C., & Kampf, S. K. (2020). Subannual Streamflow Responses to Rainfall and Snowmelt Inputs in Snow-Dominated Watersheds of the Western United States. *Water Resources Research*, 56(4), e2019WR026132.
- Havenith, H. B., Strom, A., Torgoev, I., Torgoev, A., Lamair, L., Ischuk, A., & Abdrakhmatov, K. (2015a). Tien Shan geohazards database: Earthquakes and landslides. *Geomorphology*, 249, 16-31.
- Havenith, H. B., Torgoev, A., Schlögel, R., Braun, A., Torgoev, I., & Ischuk, A. (2015b). Tien Shan geohazards database: Landslide susceptibility analysis. *Geomorphology*, *249*, 32-43.
- Keller, S., & Atzl, A. (2014). Mapping natural hazard impacts on road infrastructure—the extreme precipitation in Baden-Württemberg, Germany, June 2013. *International Journal of Disaster Risk Science*, 5(3), 227-241.
- Kirschbaum, D., Stanley, T., & Zhou, Y. (2015). Spatial and temporal analysis of a global landslide catalog. *Geomorphology*, 249, 4-15.
- Kuhn, M., & Johnson, K. (2013). Applied predictive modeling (Vol. 26). New York: Springer.
- Krøgli, I. K., Devoli, G., Colleuille, H., Boje, S., Sund, M., & Engen, I. K. (2018). The Norwegian forecasting and warning service for rainfall-and snowmelt-induced landslides. *Natural hazards* and earth system sciences, 18(5), 1427-1450.
- Giannecchini, R., Galanti, Y., Avanzi, G. D. A., & Barsanti, M. (2016). Probabilistic rainfall thresholds for triggering debris flows in a human-modified landscape. *Geomorphology*, 257, 94-107.
- Gill, J. C., & Malamud, B. D. (2014). Reviewing and visualizing the interactions of natural hazards. *Reviews of Geophysics*, 52(4), 680-722.
- Guzzetti, F., Peruccacci, S., Rossi, M., & Stark, C. P. (2007). Rainfall thresholds for the initiation of landslides in central and southern Europe. *Meteorology and atmospheric physics*, *98*(3), 239-267.
- Leonarduzzi, E., & Molnar, P. (2020). Deriving rainfall thresholds for landsliding at the regional scale: daily and hourly resolutions, normalisation, and antecedent rainfall. *Natural Hazards and Earth System Sciences*, 20(11), 2905-2919.
- Leonarduzzi, E., Molnar, P., & McArdell, B. W. (2017). Predictive performance of rainfall thresholds for shallow landslides in Switzerland from gridded daily data. *Water Resources Research*, *53*(8), 6612-6625.
- Malet, J. P., Glade, T., & Casagli, N. (Eds.). (2010). *Mountain risks: bringing science to society*. CERG Editions.
- Nikolopoulos, E. I., Borga, M., Creutin, J. D., & Marra, F. (2015). Estimation of debris flow triggering

rainfall: Influence of rain gauge density and interpolation methods. Geomorphology, 243, 40-50.

- Penna, D., Van Meerveld, H. J., Zuecco, G., Dalla Fontana, G., & Borga, M. (2016). Hydrological response of an Alpine catchment to rainfall and snowmelt events. *Journal of Hydrology*, 537, 382-397.
- Segoni, S., Piciullo, L., & Gariano, S. L. (2018). A review of the recent literature on rainfall thresholds for landslide occurrence. *Landslides*, *15*(8), 1483-1501.
- Theilen-Willige, B. (2016). Natural hazard assessment and monitoring in the Black Hills and adjacent areas, South Dakota and Wyoming, USA, Using Remote Sensing and GIS-Methods. J. Geogr. Environ. Earth Sci. Int, 1-24.

Table 1. Calibrated thresholds of mean intensity  $I_{mean}$  (mm d-1), maximum intensity  $I_{max}$  (mm d-1), and accumulated amount Q (mm) for **entire events** of the sum of rainfall and snowmelt (rainfall+snowmelt), and corresponding performance statistics for different categories of landslide size. n refers to the number of landslides in each category.

| Landslide
size   | property          | threshold | HR   | FAR  | d    | PSS  | AUC  |
|---------------------|-------------------|-----------|------|------|------|------|------|
| Small $(n = 5)$     | I mean | 10.05     | 1.00 | 0.06 | 0.06 | 0.94 | 0.96 |
|                     | I max  | 22.4      | 1.00 | 0.07 | 0.07 | 0.93 | 0.98 |
|                     | Q                 | 218.35    | 1.00 | 0.03 | 0.03 | 0.97 | 0.99 |
| Medium
(n = 41)  | I mean | 4.75      | 0.71 | 0.25 | 0.39 | 0.45 | 0.74 |
|                     | I max  | 6.75      | 0.76 | 0.30 | 0.39 | 0.46 | 0.78 |
|                     | Q                 | 9.05      | 0.83 | 0.36 | 0.40 | 0.47 | 0.79 |
| Large
(n = 11)   | I mean | 8.10      | 0.55 | 0.11 | 0.47 | 0.44 | 0.76 |
|                     | I max  | 21.75     | 0.55 | 0.05 | 0.46 | 0.49 | 0.77 |
|                     | Q                 | 75.60     | 0.55 | 0.04 | 0.46 | 0.51 | 0.77 |
| Unknown
(n = 30) | I mean | 4.05      | 0.83 | 0.34 | 0.38 | 0.49 | 0.80 |
|                     | I max  | 13.25     | 0.77 | 0.17 | 0.29 | 0.60 | 0.83 |
|                     | Q                 | 17.15     | 0.83 | 0.25 | 0.30 | 0.58 | 0.83 |

Table 2. Calibrated thresholds of mean intensity  $I_{mean}$  (mm d-1), maximum intensity  $I_{max}$  (mm d-1), and accumulated amount Q (mm) for **UTL events** of the sum of rainfall and snowmelt (rainfall+snowmelt), and corresponding performance statistics for different categories of landslide size. n refers to the number of landslides in each category.

| Landslide
size   | property          | threshold | HR   | FAR  | d    | PSS  | AUC  |
|---------------------|-------------------|-----------|------|------|------|------|------|
| Small $(n = 5)$     | I mean | 9.85      | 1.00 | 0.07 | 0.07 | 0.93 | 0.97 |
|                     | I max  | 21.55     | 1.00 | 0.07 | 0.07 | 0.93 | 0.97 |
|                     | Q                 | 124.25    | 1.00 | 0.04 | 0.04 | 0.96 | 0.98 |
| Medium
(n = 41)  | I mean | 4.80      | 0.63 | 0.25 | 0.44 | 0.39 | 0.71 |
|                     | I max  | 14.05     | 0.49 | 0.12 | 0.53 | 0.37 | 0.73 |
|                     | Q                 | 9.65      | 0.73 | 0.35 | 0.44 | 0.38 | 0.72 |
| Large
(n = 11)   | I mean | 8.10      | 0.55 | 0.11 | 0.47 | 0.44 | 0.72 |
|                     | I max  | 21.75     | 0.45 | 0.05 | 0.55 | 0.40 | 0.73 |
|                     | Q                 | 2.85      | 1.00 | 0.63 | 0.63 | 0.37 | 0.73 |
| Unknown
(n = 30) | I mean | 5.25      | 0.77 | 0.26 | 0.35 | 0.51 | 0.80 |
|                     | I max  | 13.25     | 0.73 | 0.17 | 0.32 | 0.57 | 0.81 |
|                     | Q                 | 16.90     | 0.77 | 0.25 | 0.34 | 0.51 | 0.79 |

Figure 1. Event-based temporal process of rainfall and snowmelt for selected landslide events with landslide triggers defined as (a) "rainfall", (b) "snowmelt"; (c) "mixed"; and (d) "not detected" according to the method described in section 2.2.1 in the manuscript.

---

## Author Response (AR2)

We would like to thank Referee #1 for the further time and effort spent on reviewing our manuscript. We have revised the manuscript to address the Referee's concerns. In the following, we provide point-by-point responses.

**Referee #1**

The landslide initiation/threshold should be analyzed based on solid and reliable landslide inventories obtained from field investigation or remote sensing technologies. However the landslide record used in this study extracted from GLC and GFLD mainly based on media report (more concentrated in the populated area). In my opinion, the uncertainty of imprecise landslide location could highly restrict the representative of obtained thresholds. Especially the threshold of rainfall or snowmelt was evaluated by overlaying the gridded atmospheric data, it could be direct affected by inaccurate landslide location.

AR: There are, unfortunately, no landslide inventories obtained from field investigation or remote sensing technologies with the exact landslide dates available in our study region. According to Segoni et al. (2018), who reviewed recent papers on rainfall thresholds for landslide occurrence, media report (27%) is the second major source of landslide information. Media reports used in GLC and GFLD have the advantage that they can provide accurate event dates, which enables the connecting of landslides to triggering conditions. But most media reports just record the location affected by landslides rather than the landslide initiation point. This problem also exists for databases derived from other sources, e.g., the Norwegian slide database (Jaedicke et al., 2009) primarily derived from road and railway authorities, which recorded usually the locations of deposition areas but not the initiation locations (Meyer et al., 2012). Developing landslide inventories with both high location accuracy and timing accuracy in Kyrgyzstan and Tajikistan should definitely be the focus of future studies to reduce the uncertainties in the obtained triggering thresholds.

Despite the known limitations of GLC and GFLD, they have been successfully applied in several studies for global and regional landslide assessment (e.g., Kirschbaum and Stanley, 2018; Jia et al., 2020; Stanley et al., 2020; Hunt and Dimri, 2021). Our study also demonstrates that although with unavoidable uncertainties, GLC and GFLD combined with dynamical downscaling products can distinguish atmospheric triggering conditions for landslides.

The uncertainty of imprecise landslide location perhaps can explain why numbers of historical landslides located in the zone with low to moderate susceptibility (Figure 8 of revised manuscript).

AR: Regarding this issue, we have added the following discussion in the revised manuscript:

"Around 23% of landslide events are located in zones with low and very low susceptibility. Landslide locations with low susceptibility exhibit in the eastern and southern rims of the Fergana Basin a high climatic disposition (Fig. 6). This discrepancy between the non-climatic landslide susceptibility and our mean annual exceedance maps suggests that both climatic and non-climatic aspects need to be considered for landslide susceptibility mapping. Some event locations show both low susceptibility and low climatic disposition (e.g., in southwestern Tajikistan), which implies the uncertainty in reported landslide locations."

**References:**

Hunt, K. M., & Dimri, A. P. (2021). Synoptic-scale precursors of landslides in the western Himalaya and Karakoram. *Science of the total environment*, 776, 145895.

Jia, G., Tang, Q., & Xu, X. (2020). Evaluating the performances of satellite-based rainfall data for global rainfall-induced landslide warnings. *Landslides*, *17*(2), 283-299.

Kirschbaum, D., & Stanley, T. (2018). Satellite‑based assessment of rainfall‑triggered landslide hazard for situational awareness. *Earth's future*, *6*(3), 505-523.

Meyer, N. K., Dyrrdal, A. V., Frauenfelder, R., Etzelmüller, B., & Nadim, F. (2012). Hydrometeorological threshold conditions for debris flow initiation in Norway. *Natural Hazards and Earth System Sciences*, *12*(10), 3059-3073.

Stanley, T., Kirschbaum, D. B., Pascale, S., & Kapnick, S. (2020). Extreme Precipitation in the Himalayan Landslide Hotspot. In *Satellite Precipitation Measurement* (pp. 1087-1111). Springer, Cham.

Segoni, S., Piciullo, L., & Gariano, S. L. (2018). A review of the recent literature on rainfall thresholds for landslide occurrence. *Landslides*, *15*(8), 1483-1501.